# Contrastive Symbolic Regression: Aligned Representations, Adaptive Prediction, and Diverse Ensembles

Hengzhe Zhang [1]   Qi Chen [1]   Bing Xue [1]   Wolfgang Banzhaf [2]   Mengjie Zhang [1]

## Abstract

Existing symbolic regression approaches primarily focus on learning explicit input-output mappings, often neglecting relational structures among data instances. This paper introduces Contrastive Symbolic Regression (CSR), a feature-construction-based symbolic regression approach that integrates evolutionary feature construction with contrastive learning to shape a representation space where geometric proximity reflects similarity in the target space. CSR employs a contrastive objective that optimizes a linear transformation of constructed features, with a closed-form solution for aligning the feature space with the target space. The constructed features are applied to K-nearest neighbor regression, where we propose an efficient leave-one-out cross-validation (LOOCV) method that addresses standard LOOCV's computational expense and adaptively selects the neighborhood size, along with a linear-rank weighted K-nearest neighbor variant for faithful assessment of representation quality during evolution. A determinant point process-based ensemble selection mechanism further enhances robustness by jointly considering model quality and diversity. Extensive experiments on 58 real-world regression datasets demonstrate that CSR consistently surpasses both traditional symbolic regression and modern machine learning counterparts, highlighting CSR as a promising direction for interpretable and effective regression modeling.

[1]Centre for Data Science and Artificial Intelligence & School of Engineering and Computer Science, Victoria University of Wellington, PO Box 600, Wellington 6140, New Zealand [2]Department of Computer Science and Engineering, Michigan State University, East Lansing, MI 48824, USA. Correspondence to: Qi Chen <qi.chen@ecs.vuw.ac.nz>.

*Proceedings of the 43$^{rd}$ International Conference on Machine Learning*, Seoul, South Korea. PMLR 306, 2026. Copyright 2026 by the author(s).

## 1. Introduction

Symbolic regression (SR) aims to discover interpretable mathematical expressions that capture relationships in data (Shojaee et al., 2025). Unlike deep learning-based regression with large numbers of parameters that learn direct input-output mappings $f : X \mapsto y$, symbolic regression searches over symbolic expressions defined by an operator set $\mathcal{O}$ and a terminal set $\mathcal{T}$, seeking formulas that balance predictive accuracy with interpretability and simplicity. However, SR methods often lack an explicit mechanism to align geometric proximity in the feature space with semantic similarity in the continuous target space, making it difficult to capture local patterns and non-parametric variation.

A prominent paradigm within SR is feature construction–based symbolic regression (Cava et al., 2019; Fong & Motani, 2025), which evolves composite features $\Phi(X) = [\phi_1(X), \dots, \phi_m(X)]$ for downstream learners rather than modeling $f(X)$. This decoupling allows symbolic regression to leverage classical interpretable machine learning algorithms to improve predictive performance. Existing studies primarily combine symbolic features with linear regression, where evolved features act as basis functions for parametric fitting. However, real-world datasets often exhibit both global parametric trends, which are smooth and predictable patterns described by mathematical equations, and local non-parametric variations, which reflect complex localized and data-driven behaviors. This combination motivates hybrid approaches that integrate ordinary symbolic regression for direct parametric mappings with feature construction methods that capture rich local patterns through relational structures among data instances.

To capture local patterns, the feature map should encode relational structures such that geometric proximity reflects semantic similarity in the target space. Contrastive learning can achieve this by optimizing pairwise relationships (Chen et al., 2020), bringing similar instances together while pushing dissimilar ones apart. This alignment shapes the feature space geometry to match the data structure, benefiting distance-based methods such as K-nearest neighbor (KNN) regression and making it a natural direction to extend existing feature construction–style symbolic regression. This alignment is particularly appealing in applications such as

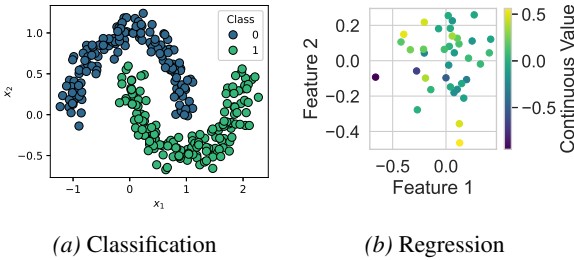

*(a)* Classification        *(b)* Regression

*Figure 1.* Illustration of the challenge in representation learning for regression.

loan approval, where decisions should be justified by similar past instances (Yu et al., 2024). KNN can support a prediction by retrieving similar historical instances from the learned symbolic feature space, thereby providing instance-level interpretability.

However, integrating contrastive learning with feature construction–based symbolic regression remains unexplored. The core challenge is that standard contrastive objectives often rely on binary or ordinal positive/negative relations, whereas regression involves continuous target values. Unlike classification (La Cava et al., 2019b) or clustering (Lensen et al., 2020), as illustrated in Figure 1a, regression requires distances to reflect similarities in target values. As illustrated in Figure 1b, this requires constructing a feature space whose geometry reflects the continuous structure of the target values. Ordinal contrastive formulations preserve ranked label relationships (Baek et al., 2024), but regression involves continuous supervision, and reducing targets to ordinal relations discards exact magnitude information. Ideally, the feature space geometry should reflect continuous target proximity.

Genetic Programming (GP), a symbolic, population-based search method, provides strong feature construction capabilities, but integrating contrastive learning within evolution presents several challenges. First, GP excels at structural exploration but struggles with precise coefficient optimization, which is crucial for maintaining target–distance consistency in regression. Second, while many contrastive objectives are optimized with iterative gradient-based methods (Chen et al., 2020), integrating such optimization within evolution is computationally expensive, motivating a closed-form solution. Third, when using distance-based methods, e.g., KNN, as the downstream learner, two practical challenges must be addressed: sensitivity to the neighborhood size $K$ and uniform averaging of neighbors that fails to capture local geometric structure. Fourth, the high variance typical of GP-based feature construction (Owen et al., 2020) necessitates robust ensemble selection.

To address these limitations, we propose Contrastive Symbolic Regression[1], an efficient framework that integrates evolutionary feature construction with contrastive learning. The framework constructs a feature space where geometric distances reflect the structure of the target space, focusing on relational structures rather than direct feature-to-target mappings. This learned representation is used in a boosting base learner, where ridge regression first captures the global linear trend and KNN then models local residual patterns, with efficient evaluations and reduced GP variance through robust ensemble modeling. The key contributions are:

- We derive a closed-form solution for a contrastive transformation matrix $W$ that projects GP-constructed features into a space where geometric distances reflect the structure of the target space, enabling efficient optimization without gradient-based methods.

- We develop a KNN-based evaluation framework for assessing constructed feature spaces, including an efficient leave-one-out cross-validation (LOOCV) method for selecting the optimal neighborhood size $K$ and a linear-rank weighted KNN evaluation that provides a more faithful measure of representation quality.

- We introduce a determinantal point process (DPP)-based ensemble selection mechanism that reduces symbolic regression variance by promoting diverse high-quality individuals, leading to more robust ensemble models.

## 2. Related Work

Recent advances in symbolic regression fall into two categories. Search-based methods, such as Genetic Programming (GP) (Jiang & Xue, 2024; Fong et al., 2023) and Monte Carlo Tree Search (MCTS) (Kamienny et al., 2023; Sun et al., 2023), explicitly explore the symbolic expression space through evolutionary or probabilistic search, offering strong interpretability and structural exploration capabilities. Learning-based methods instead optimize differentiable or data-driven representations: reinforcement learning formulates symbolic regression as sequential program generation (Landajuela et al., 2021; Xu et al., 2024b), sparse neural networks approximate symbolic operators with sparsity constraints (Sahoo et al., 2018), and foundation models leverage large pretrained architectures for equation generation (Shojaee et al., 2025; Meidani et al., 2024; Grayeli et al., 2024), improving scalability and learning efficiency. Hybrids of these approaches (Landajuela et al., 2022; Shojaee et al., 2024; Grayeli et al., 2024) have further improved the scalability and flexibility of symbolic regression. However, a fundamental limitation shared by all these approaches is that they learn direct mappings from features to targets

---
[1]Source code: https://github.com/hengzhe-zhang/CSR-ICML/

$f : X \mapsto y$, focusing on global parametric relationships while often neglecting relational structures among data instances that are crucial for capturing local patterns and non-parametric variations in regression tasks.

## 3. Methodology

### 3.1. Algorithm Framework

This work proposes an efficient and robust symbolic regression framework consisting of three components: contrastive learning that aligns feature space geometry with the target space, adaptive KNN regression that uses the constructed features, and DPP-based ensemble selection that maintains diverse high-quality models. The overall evolutionary feature construction process is illustrated in Figure 2. The process initializes a population of symbolic trees using sparse neural network-based and traditional GP methods. During solution evaluation, two transformation processes occur: first, GP feature construction creates features from symbolic trees; second, contrastive learning, described in Section 3.2, transforms the GP-constructed features into a target-aligned space. The transformed features feed a boosting base learner where ridge regression captures the global linear trend and KNN models the local residual patterns to obtain the prediction error. Parent selection uses double lexicase selection to encourage concise expressions. Offspring are generated through random subtree crossover and mutation. Finally, DPP-based ensemble selection, described in Section 3.4, maintains an archive of diverse high-quality individuals for robust ensemble predictions. The detailed algorithmic description is provided in Appendix B.

### 3.2. Contrastive Symbolic Regression

The key idea of Contrastive Symbolic Regression (CSR) is to move beyond fitting a parametric mapping $f : X \mapsto y$ from features to targets, and instead learn the relationships among instances in a constructed feature space. Specifically, given two data points $x_i$ and $x_j$ with targets $y_i$ and $y_j$, if $|y_i - y_j|$ is small, their symbolic representations $\Phi(x_i)$ and $\Phi(x_j)$ should be mapped close together:

$$\|\Phi(x_i) - \Phi(x_j)\|^2 \;\; \propto \;\; |y_i - y_j|^2. \tag{1}$$

This semantics-driven search perspective transforms symbolic regression into a representation learning problem, where predictions are obtained by retrieving and aggregating information from nearest neighbors in the learned feature space.

#### 3.2.1. MODEL AND PREDICTION

Each solution is represented by $m$ GP trees, which construct $m$ new features from the original data. This yields the feature space

$$\Phi(X) = [\phi_1(X), \ldots, \phi_m(X)] \in \mathbb{R}^{n \times m}, \tag{2}$$

where $n$ is the number of samples. Predictions for a new input instance $x_j$ are obtained by first mapping it into the transformed feature space $z_j = W^\top \Phi(x_j)$, where $W \in \mathbb{R}^{m \times m}$ is a linear transformation matrix, and then averaging the targets of its $K$-nearest neighbors: $\hat{y}_j = \frac{1}{K} \sum_{i \in \mathcal{N}_K(z_j)} y_i$, where $\mathcal{N}_K(z_j)$ denotes the set of $K$-nearest neighbors of $z_j$ in the transformed space.

#### 3.2.2. TRAINING WITH CONTRASTIVE LEARNING

To ensure that the transformed space aligns with the target space, we introduce a target-based distance matrix $D \in \mathbb{R}^{n \times n}$ with entries $D_{ij} = (y_i - y_j)^2$. For the transformed features, the corresponding distance matrix is $D' \in \mathbb{R}^{n \times n}$ with entries $D'_{ij} = \|W^\top \Phi(x_i) - W^\top \Phi(x_j)\|^2$. The weight matrix $W \in \mathbb{R}^{m \times m}$ is optimized by minimizing the difference between the feature and target distance matrices:

$$\arg\min_W \sum_{i,j} \left( \|W^\top \Phi(x_i) - W^\top \Phi(x_j)\|^2 - (y_i - y_j)^2 \right)^2. \tag{3}$$

This contrastive objective enforces that samples with similar target values are closer in the transformed feature space, while dissimilar samples are pushed apart. In contrast to ordinal contrastive losses that define ordered positive/negative relations and require gradient-based optimization (Baek et al., 2024), CSR preserves relationships based on continuous target proximity by directly matching pairwise target distances, which yields the closed-form solution in Equation (6). For large datasets, constructing and manipulating the full matrix $D$ is expensive. To address this, we adopt random subsampling: when the dataset size exceeds a predefined threshold $S$, a random subset of $S$ samples is selected to compute $W$. Note that subsampling is applied only during the optimization of $W$; after contrastive training, the contrastively transformed features $z$ are used for regression on the full dataset.

This alignment of $D' \approx D$ benefits downstream regression: for any query $x_i$, its neighbors $j \in \mathcal{N}_K(z_i)$ in the transformed space are more likely to have small target differences $|y_i - y_j|$, improving prediction accuracy. The contrastive objective also regularizes the feature map by suppressing feature dimensions uncorrelated with target structure, yielding more generalizable embeddings.

#### 3.2.3. CLOSED-FORM SOLUTION

We aim to learn a transformation matrix $W \in \mathbb{R}^{m \times m}$ such that the transformed pairwise distance matrix $D'$ approximates the target distance matrix $D$. Based on the con-

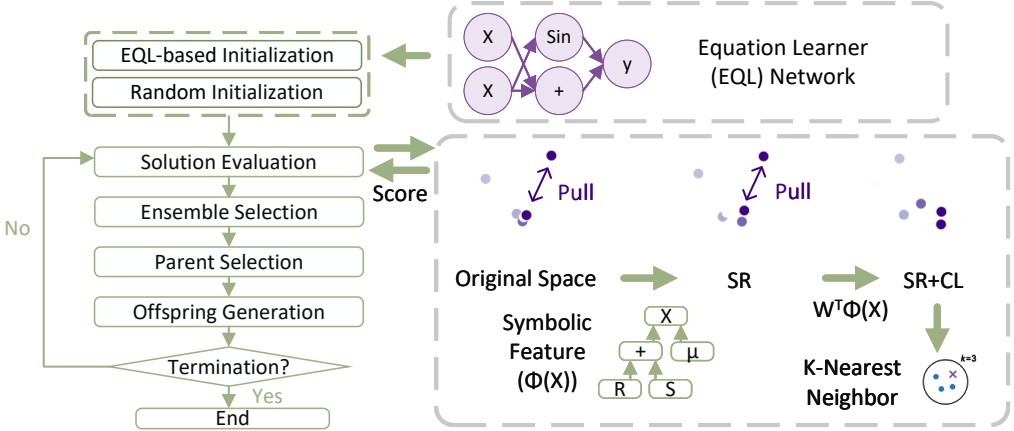

*Figure 2.* Workflow of Contrastive Symbolic Regression.

trastive objective in Equation (3), expanding $D'$ and vectorizing over all pairs results in a least-squares problem over $\mathbf{v} = \mathrm{vec}(M)$, where $M = WW^\top$. Introducing an $\ell_2$ regularization term gives the ridge-regularized objective:

$$\mathcal{L} = \|d - B\mathbf{v}\|^2 + \lambda\|\mathbf{v}\|^2, \qquad (4)$$

where $d$ is the vectorized target distance matrix and $B$ encodes pairwise feature differences. The full derivation is provided in Appendix J. The closed-form ridge solution is:

$$\mathbf{v}^* = (B^\top B + \lambda I)^{-1}B^\top d. \qquad (5)$$

To recover the transformation matrix $W$, we reshape $\mathbf{v}^*$ into $M$, symmetrize it as $M \leftarrow (M + M^\top)/2$, and then perform an eigendecomposition:

$$M = U\Lambda U^\top, \quad W = U\Lambda_+^{1/2}, \qquad (6)$$

where $U$ is the matrix of eigenvectors and $\Lambda$ is the diagonal matrix of eigenvalues. We form $\Lambda_+$ by replacing negative eigenvalues in $\Lambda$ with zero before taking the square root. This analytical formulation provides an efficient and stable solution for aligning feature and target distances without iterative gradient-based optimization.

### 3.2.4. LINEAR REGRESSION BOOSTING

Contrastive learning is inherently non-parametric; it models complex instance-level relationships without assuming a specific functional form, but it may underperform when the target function contains a strong parametric component. For example, explicit functional modeling is better at interpolation in regions where samples are sparse. To improve generalization, we combine contrastive learning with ridge regression through a boosting strategy: a ridge regression model is first fitted to capture the global linear structure of the data in the constructed feature space $\Phi(X)$, and contrastive learning is then applied to model the residuals.

Formally, ridge regression produces predictions $\hat{y}_{\mathrm{ridge}} = \Phi(X)\beta$, where the coefficient vector $\beta$ is obtained by minimizing the regularized loss

$$\beta = \arg\min_\beta \|y - \Phi(X)\beta\|^2 + \lambda\|\beta\|^2. \qquad (7)$$

We then compute the residual $e = y - \hat{y}_{\mathrm{ridge}}$ and train the contrastive learning component on the residuals $e$, producing predictions $\hat{e} = f_{\mathrm{KNN}}(\Phi(X)W)$. The final boosted prediction is obtained by combining the two components: $\hat{y} = \hat{y}_{\mathrm{ridge}} + \hat{e}$. This two-stage framework allows ridge regression to explain global trends, while contrastive learning focuses on capturing complex, nonlinear residual relationships. It is important to note that this two-stage approach is only applied within the solution evaluation stage. The symbolic features are evolved based on holistic fitness, which means that constructed features that are not perfect on each model individually but have synergistic effects are also considered good individuals.

### 3.3. Evaluating Feature Space via KNN Regression

The aim of the constructed feature space is to enhance the predictive performance of KNN regression. However, evaluating features using training loss on KNN can lead to overfitting, and KNN performance is sensitive to the choice of neighborhood size $K$. Standard leave-one-out cross-validation (LOOCV) is expensive, requiring $n$ separate model fits. To address this, we propose an efficient LOOCV approach for selecting the optimal $K$ with computational cost comparable to fitting a single KNN. This LOOCV-based selection, however, has limitations when evaluating feature space quality. To overcome these, we propose a linear-rank weighted KNN regression that better captures local structure and provides a more faithful measure of representation quality.

**Algorithm 1** Efficient LOOCV-Based $K$ Selection for KNN

1: idx ← Neighbors($X, K_{\max}+1$)  ▷ Query nearest neighbors
2: idx ← idx[:, 2:$K_{\max}+1$]                    ▷ Exclude self
3: $\mathbf{Y}_{\text{nn}}$ ← $y$[idx]
4: $\mathbf{S}_{\text{cum}}$ ← cumsum($\mathbf{Y}_{\text{nn}}$)       ▷ Prefix sums over neighbors
5: $\mathbf{M}_{:,k}$ ← $\mathbf{S}_{\text{cum},:,k}/k$            ▷ Running means
6: $\mathbf{E}_{:,k}$ ← $(y - \mathbf{M}_{:,k})^2$            ▷ Squared errors
7: $R^2_{\text{LOO}}$ ← $1 - \dfrac{\text{mean}(\mathbf{E}, \text{axis} = 0)}{\text{Var}(y)}$
8: $K^\star$ ← $\arg\max_{K_{\min} \le k \le K_{\max}} R^2_{\text{LOO}}[k]$
9: **return** $K^\star$

### 3.3.1. EFFICIENT LOOCV-BASED $K$ SELECTION

The predictive performance of KNN regression depends critically on the neighborhood size $K$. Standard LOOCV requires fitting $n$ separate models, one per left-out sample, making exhaustive $K$ search expensive. In contrast, we introduce a fully vectorized LOOCV procedure that evaluates all candidate $K$ values simultaneously in a single nearest-neighbor retrieval.[2] This reduces LOOCV from $O(nK_{\max})$ evaluations to the cost of one KNN query, where $K_{\max}$ is the maximum number of neighbors.

Given training data $\mathcal{D} = \{(x_i, y_i)\}_{i=1}^n$, we retrieve $K_{\max} + 1$ nearest neighbors for each sample, discard the nearest neighbor, which is essentially the sample itself, and collect target values into $\mathbf{Y}_{\text{nn}} \in \mathbb{R}^{n \times K_{\max}}$. Cumulative mean predictions $\hat{y}_i^{(k)} = \frac{1}{k} \sum_{j=1}^k y_{i,j}$ are computed for all $k \in [1, K_{\max}]$, and the best $K^\star = \arg\max_{K_{\min} \le k \le K_{\max}} R^2_{\text{LOO}}(k)$ is selected, where

$$R^2_{\text{LOO}}(k) = 1 - \frac{\frac{1}{n} \sum_{i=1}^n (y_i - \hat{y}_i^{(k)})^2}{\text{Var}(y)}. \tag{8}$$

The full procedure is detailed in Algorithm 1. This vectorized approach computes all $R^2_{\text{LOO}}(k)$ values in a single pass and is applied at the end of evolution to determine the optimal $K$ for each ensemble model.

### 3.3.2. LINEAR-RANK WEIGHTED KNN EVALUATION

Although adaptive selection of $K$ is suitable for KNN learning, it is unsuitable for evaluation. The core issue is that LOOCV evaluates a predictor and a hyperparameter jointly: the resulting score mixes representation quality with the choice of $K$, making fitness scores across individuals not directly comparable. For example, one feature space may select $K=2$ while another selects $K=10$; since $K$ heavily influences prediction quality, these scores reflect different model configurations rather than feature space quality alone. To address this, we propose to use rank-weighted KNN with a fixed $K$ to evaluate feature space quality. We extend standard KNN with linear-rank weighting: instead of uniform averaging, we assign linearly decreasing

weights $w_r = \frac{2(K-r+1)}{K(K+1)}$ to neighbors ranked by distance, where $r = 1$ is the nearest neighbor. The prediction is $\hat{y} = \sum_{r=1}^K w_r y_r$, emphasizing closer neighbors and better reflecting geometric consistency. After feature evolution completes, LOOCV-based $K$ selection is applied to determine the optimal neighborhood size for each model in the final ensemble.

### 3.4. DPP-Based Ensemble Selection

To improve generalization, a determinantal point process (DPP)-based ensemble selection mechanism enhances diversity and reduces redundancy among high-fitness GP individuals. DPPs define probability distributions over subsets of individuals that balance quality and diversity. Intuitively, subsets containing high-quality and mutually dissimilar individuals receive higher probabilities, encouraging the selection of individuals that are both accurate and complementary in behavior. Formally, for an ensemble subset $\mathcal{S}$ of a finite candidate set $\mathcal{C}$, the probability follows $\mathbb{P}(\mathcal{S}) \propto \det(L_\mathcal{S})$, where $L \in \mathbb{R}^{|\mathcal{C}| \times |\mathcal{C}|}$ is a kernel matrix encoding pairwise similarities among the $|\mathcal{C}|$ candidates. The determinant $\det(L_\mathcal{S})$ measures the volume spanned by the semantics/output vectors of individuals in $\mathcal{S}$: larger volumes correspond to more diverse subsets.

### 3.4.1. KERNEL CONSTRUCTION AND FORMULATION

The DPP kernel captures individual quality and similarity, allowing the selection of strong yet non-redundant GP individuals. Let $\mathcal{C}$ denote the candidate set for selection, and let $y \in \mathbb{R}^n$ denote the target outputs for $n$ training instances. Each candidate $c_i \in \mathcal{C}$ produces a semantic prediction $f_{c_i}(\Phi_{c_i}(X)) \in \mathbb{R}^n$, where $\Phi_{c_i}$ is the learned feature mapping. Stacking all predictions yields $F \in \mathbb{R}^{|\mathcal{C}| \times n}$, and centering semantics relative to the target outputs gives $R = F - \mathbf{1}_{|\mathcal{C}|} y^\top$, where $\mathbf{1}_{|\mathcal{C}|}$ is an all-ones column vector. Each row $R_i \in \mathbb{R}^n$ represents the residual semantics of candidate $c_i$.

The normalized fitness of candidate $c_i$ is $q_i = (f_i - f_{\min})/(f_{\max} - f_{\min} + \epsilon)$, and the quality vector is $\mathbf{q} = [q_1, q_2, \ldots, q_{|\mathcal{C}|}]^\top \in \mathbb{R}^{|\mathcal{C}|}$. The DPP kernel $L \in \mathbb{R}^{|\mathcal{C}| \times |\mathcal{C}|}$ combines fitness and pairwise similarity as $L_{ij} = q_i q_j \text{sim}(R_i, R_j)$, where $\text{sim}(R_i, R_j) = R_i^\top R_j / (\|R_i\| \|R_j\|)$ denotes cosine similarity. The term $q_i q_j$ favors strong candidates, while the similarity term suppresses redundancy among semantically similar ones.

The cosine similarity kernel is used because it emphasizes angular relationships among semantic vectors, producing more meaningful diversity among high-performing models, while the Euclidean RBF kernel may introduce spurious diversity from low-fitness individuals, as shown in Appendix O in the appendix.

---

[2]For comparisons with non-vectorized LOOCV (Kanagawa, 2024), see Appendix C.

**Algorithm 2** DPP-Based Ensemble Selection

---

**Require:** Population $P$, targets $y \in \mathbb{R}^n$, archive $A$, ensemble size $E$
**Ensure:** Updated archive $A$
1: $\mathcal{C} \leftarrow \text{Top-}E(P) \cup A$
2: $R \leftarrow [f_c(\Phi_c(X)) - y]_{c \in \mathcal{C}}$
3: $\mathbf{q} \leftarrow [(f_i - f_{\min})/(f_{\max} - f_{\min} + \epsilon)]_{i=1}^{|\mathcal{C}|}$
4: $L_{ij} \leftarrow q_i q_j \dfrac{R_i^\top R_j}{\|R_i\|\|R_j\|}$        ▷ Cosine kernel
5: $\mathcal{S} \leftarrow \emptyset$
6: **for** $t = 1$ to $E$ **do**
7:     $i^* \leftarrow \arg\max_i [L_{ii} - L_{i\mathcal{S}}(L_{\mathcal{S}\mathcal{S}} + \epsilon I)^{-1} L_{\mathcal{S}i}]$ ▷ Marginal gain
8:     $\mathcal{S} \leftarrow \mathcal{S} \cup \{i^*\}$
9: **end for**
10: $A \leftarrow \{\mathcal{C}[i] \mid i \in \mathcal{S}\}$        ▷ Archive update

---

### 3.4.2. GREEDY SELECTION

Exact DPP sampling is expensive; therefore, a greedy maximum a posteriori (MAP) approximation (Chen et al., 2018) is employed. Starting with an empty set $\mathcal{S} = \emptyset$, the algorithm iteratively adds the individual maximizing the marginal gain

$$i^* = \arg\max_i \big[ L_{ii} - L_{i\mathcal{S}}(L_{\mathcal{S}\mathcal{S}} + \epsilon I)^{-1} L_{\mathcal{S}i} \big], \quad (9)$$

where $\mathcal{S}$ denotes the indices of selected individuals. The process continues until $|\mathcal{S}| = E$, where $E$ is the ensemble size. At each generation, the ensemble archive $A$ is updated by first merging the current generation's top-performing individuals with those already stored in the archive, forming a candidate set $\mathcal{C} = \text{Top-}E(P) \cup A$. DPP selection is then applied to this candidate set to retain a subset of $E$ individuals that are both high-quality and diverse. The overall procedure is summarized in Algorithm 2.

## 4. Experimental Settings

### 4.1. Datasets

Experiments are conducted on 58 real-world regression datasets from the Penn Machine Learning Benchmark (PMLB) (Olson et al., 2017). The 62 synthetic datasets generated from the Friedman function in PMLB are excluded from our experiments, as these synthetic functions are simple and insufficient to reflect the complexity of real-world datasets, which contain noise and exhibit complex nonlinear relationships.

### 4.2. Evaluation Protocol

For ablation studies, we randomly split each dataset into 80% training and 20% test sets (Holzmüller et al., 2024). To be consistent with SRBench (Cava et al., 2021), we cap the training set at 10,000 instances. All features are standardized to zero mean and unit variance. Predictive performance is measured using the coefficient of determination $R^2 = 1 - \frac{\sum_i (y_i - \hat{y}_i)^2}{\sum_i (y_i - \bar{y})^2}$, where higher values indicate better accuracy. Each experiment is repeated 30 times with different random seeds to ensure statistical reliability. The Wilcoxon signed-rank test at a 0.05 significance level is used for pairwise comparisons. The parameter settings of the proposed method and baselines are detailed in Appendix H. For comparisons with machine learning baselines, we follow the SRBench evaluation protocol (Cava et al., 2021).

### 4.3. Baseline Methods

We compare the proposed method against both symbolic regression and conventional machine learning baselines. The baselines include RealMLP (Holzmüller et al., 2024), a strong MLP for tabular data; ModernNCA (Ye et al., 2025), a deep metric learning method that modernizes neighborhood component analysis with deep representations; MLP-PLR (Rubachev et al., 2025), which models numerical features using learnable periodic embeddings; and tree-based ensemble methods such as XGBoost and LightGBM. Hyperparameters are fine-tuned on the training split using heteroscedastic Bayesian optimization (HEBO) (Cowen-Rivers et al., 2022). Details are provided in Appendix Q.

To examine the generality of the proposed framework, feature construction-style symbolic regression methods are applied to several different base learners and transformation methods. Table 1 provides an overview of the components used in each variant. Detailed descriptions of each variant are provided in Appendix M. All symbolic regression-based methods share the same evolutionary framework and hyperparameter configurations listed in Table 5 of the Appendix. Each method is evaluated under the ensemble learning setting with identical ensemble selection and size. All symbolic regression-based methods use the same DPP ensemble selection proposed in Section 3.4.

## 5. Experimental Results

### 5.1. Comparison Among Symbolic Regression Variants

**Base Learners.** Figure 3 compares the test $R^2$ of different base learners across the 58 datasets. $\text{SR}^{\text{KNN}}$ underperforms $\text{SR}^{\text{LR}}$ and $\text{SR}^{\text{LR+KNN}}$, showing that naive instance-level regression lacks expressive power. However, by incorporating contrastive learning and LOOCV-based $K$ selection, $\text{SR}^{\text{CL+LOO-KNN}}$ achieves performance competitive with $\text{SR}^{\text{LR}}$. $\text{SR}^{\text{LR+KNN}}$ models global and local patterns simultaneously and achieves better performance than using either $\text{SR}^{\text{LR}}$ or $\text{SR}^{\text{KNN}}$ alone. $\text{SR}^{\text{LR+CL+KNN}}$ achieves further gains by explicitly aligning similar samples through contrastive learning. Finally, $\text{SR}^{\text{All}}$ achieves the best performance with linear-rank weighted KNN and adaptive $K$

*Table 1.* Overview of symbolic regression variants and their components.

| Variant | LR | KNN | DT | PLS[*] | LPP[*] | CL | LOOCV $K$ |
|---|---|---|---|---|---|---|---|
| SR[LR] (La Cava & Moore, 2020) | ✓ | | | | | | |
| SR[KNN] (La Cava et al., 2019b) | | ✓ | | | | | |
| SR[CL+KNN] | | ✓ | | | | ✓ | |
| SR[DT] (Zhang et al., 2021) | | | ✓ | | | | |
| SR[LR+KNN] | ✓ | ✓ | | | | | |
| SR[LR+DT] (Zhang et al., 2023b) | ✓ | | ✓ | | | | |
| SR[LR+PLS+KNN] | ✓ | ✓ | | ✓ | | | |
| SR[LR+LPP+KNN] | ✓ | ✓ | | | ✓ | | |
| SR[LR+CL+KNN] | ✓ | ✓ | | | | ✓ | |
| SR[CL+LOO-KNN] | | ✓ | | | | ✓ | ✓ |
| SR[LOO-KNN] | | ✓ | | | | | ✓ |
| SR[All] | ✓ | ✓ | | | | ✓ | ✓ |

[*] PLS: Partial Least Squares (Wold et al., 1984), LPP: Locality Preserving Projection (He & Niyogi, 2003)

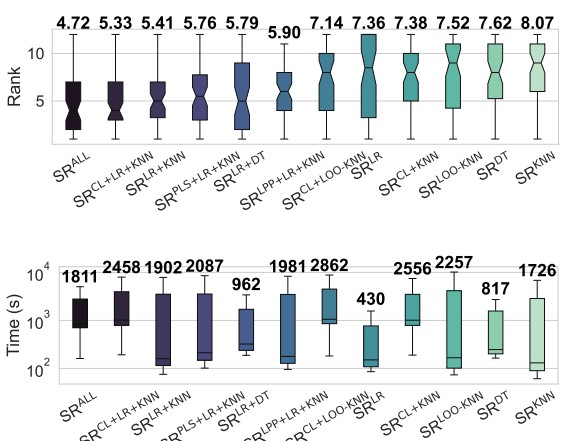

*Figure 3.* Comparisons across symbolic regression variants. Top: average rank of methods based on test $R^2$ across datasets; Bottom: Training time. Average values are shown in the figure.

selection building on SR[LR+CL+KNN].

**Learning Objectives.** Alternative transformation objectives such as SR[LR+PLS+KNN] and SR[LR+LPP+KNN] are evaluated to assess the effect of linear and locality-preserving mappings. SR[LR+CL+KNN] outperforms both, demonstrating that contrastive learning produces more discriminative features than PLS or LPP projections by directly optimizing feature space relationships according to the target space. The advantage of contrastive learning is more pronounced when using either LOOCV KNN or standard KNN as baseline learners, as the results show that SR[CL+LOO-KNN] outperforms SR[LOO-KNN] and SR[CL+KNN] outperforms SR[KNN].

Table 2 reports the normalized median $R^2$ and bootstrapped 95% confidence intervals for each variant across the 58 datasets, where the median $R^2$ is normalized per dataset to account for varying difficulty levels. These results confirm that contrastive learning provides a clear and consistent benefit over naive KNN, and that adding each component—ridge regression, LOOCV-based $K$ selection, and linear-rank weighting-based evaluation—progressively improves performance, with SR[All] achieving the best result.

*Table 2.* Normalized median $R^2$ and bootstrapped 95% confidence intervals across 58 datasets. Median $R^2$ is normalized per dataset to account for varying difficulty levels.

| Variant | Norm. Med. $R^2$ | 95% CI |
|---|---|---|
| SR[All] | 0.8884 | [0.7787, 0.9547] |
| SR[LR+CL+KNN] | 0.8310 | [0.7528, 0.8579] |
| SR[LR+PLS+KNN] | 0.8244 | [0.6766, 0.8578] |
| SR[LR+KNN] | 0.8241 | [0.7120, 0.8967] |
| SR[LR+LPP+KNN] | 0.7647 | [0.6756, 0.8392] |
| SR[LR+DT] | 0.7628 | [0.5499, 0.8615] |
| SR[LR] | 0.6602 | [0.2731, 0.7914] |
| SR[CL+LOO-KNN] | 0.6410 | [0.5307, 0.7458] |
| SR[CL+KNN] | 0.6064 | [0.4541, 0.7437] |
| SR[LOO-KNN] | 0.5905 | [0.4353, 0.6900] |
| SR[DT] | 0.4861 | [0.3597, 0.7119] |
| SR[KNN] | 0.4613 | [0.3538, 0.6254] |

### 5.2. Computational Cost Analysis

The proposed method incurs higher computational cost than SR[LR], primarily due to optimizing the transformation matrix and performing KNN predictions. This increase is acceptable in practice given the significant improvements in predictive performance reported in Figure 3. The overall runtime remains within the same order of magnitude because the transformation matrix has a closed-form analytical solution, enabling efficient optimization within each generation. Theoretically, solving Equation (23) has complexity $O(S^2 m^4)$, where $S$ is the subsampling size and $m$ is the number of constructed features. In our experiments, we set $S = 100$ and $m = 10$ to keep this component efficient. In contrast, GP-based feature construction scales as $O(nms)$, where $n$ is the number of instances and $s$ is the average size of GP trees. KNN prediction is $O(n^2)$ when evaluated for all instances. Thus, as shown in Figure 3, KNN evaluation is the major computational cost, especially on larger datasets due to its quadratic dependence on $n$.

### 5.3. Comparisons with Symbolic Regression and Machine Learning Methods

Figure 4 summarizes the $R^2$ scores, model sizes, and training times of 27 symbolic regression and ML algorithms across 58 real-world datasets from SRBench. The proposed method achieves the highest overall performance. Compared with RAG-SR (Zhang et al., 2025), the proposed method attains higher $R^2$ with comparable training time, indicating that contrastive representation learning yields stronger generalization than error-driven feature construction. Compared with modern deep learning methods such as RealMLP (Holzmüller et al., 2024), MLP-PLR (Rubachev et al., 2025), and ModernNCA (Ye et al., 2025), the proposed method achieves superior generalization due to symbolic feature construction. Moreover, the proposed method

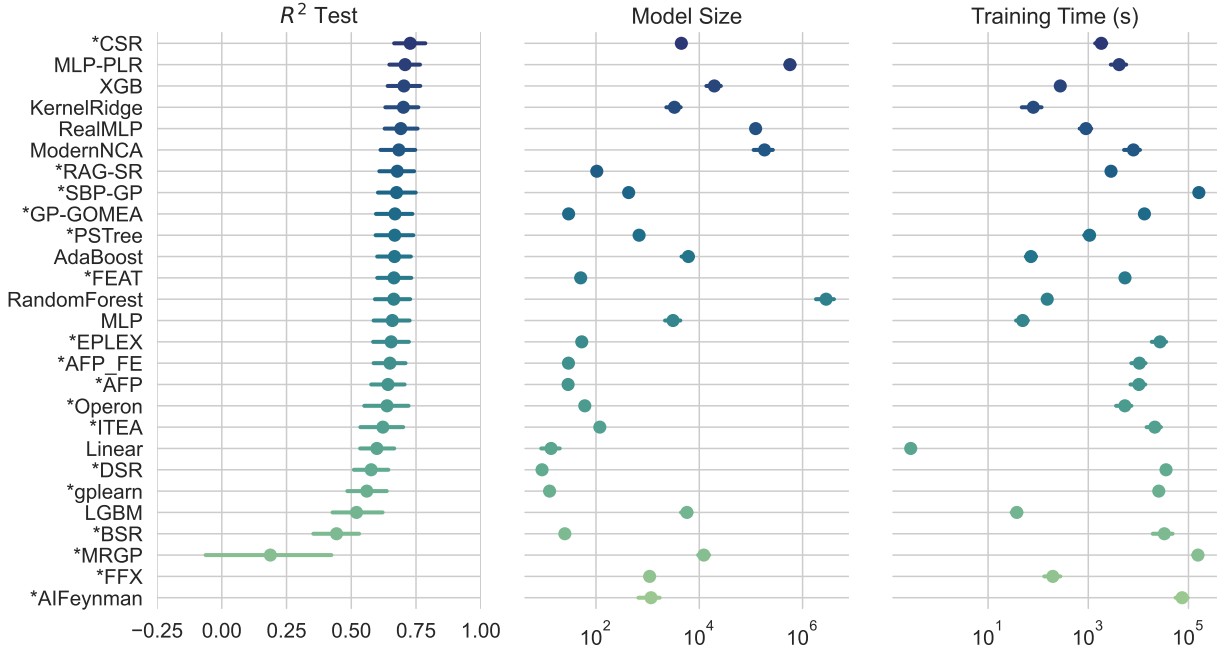

*Figure 4.* $R^2$ scores, model sizes, and training times of 27 symbolic regression and machine learning algorithms across 58 real-world datasets in the symbolic regression benchmark. The asterisk (*) denotes symbolic regression algorithms.

has comparable training time when hyperparameter tuning is considered. The proposed method also offers improved interpretability, since important symbolic features can be extracted, as illustrated in Appendix E. Even compared with highly competitive tree-based learners such as XGBoost and LightGBM, KNN with symbolic regression-based feature construction achieves superior performance while maintaining simplicity.

### 5.4. Visualization of Mapping Space

To illustrate the effect of contrastive learning, PCA is applied to project the constructed feature space into two dimensions using all training samples from the representative dataset OpenML 1089. The target values shown are residuals after ridge regression, highlighting the local patterns that need to be captured by KNN. As shown in Figure 5a, without feature construction, samples with similar target values are widely scattered without any clear structure. The constructed features shown in Figure 5b become more organized, with samples partially grouping according to similar targets. Incorporating contrastive learning, as visualized in Figure 5c, further refines the mapping, producing clearer separations among samples.

Figure 6 compares pairwise distance matrices across different feature spaces using the same setting as Figure 5. In Figure 6a, the distance matrix deviates substantially from the target in Figure 6d, explaining weak KNN performance. The constructed features in Figure 6b improve alignment but large deviations remain. After contrastive learning in Fig-

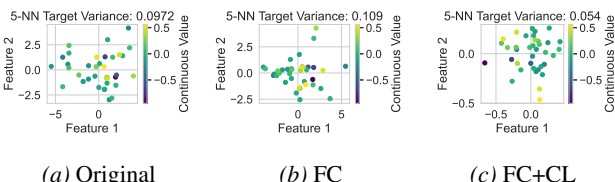

| *(a)* Original | *(b)* FC | *(c)* FC+CL |

*Figure 5.* Instance distributions in feature space after PCA projection. Each point is an instance. Color indicates target values. FC denotes feature construction.

ure 6c, the learned feature space closely matches the target distance structure, explaining improved generalization.

### 5.5. Instance-Level Interpretability

One advantage of CSR is that it supports instance-level interpretability through KNN retrieval. Figure 7 shows this for the best individual in terms of fitness of the SR[CL+LOO-KNN] variant on the US crime rate dataset. The learned symbolic space retrieves neighbors whose target values are closer to the test target than those found in the original feature space. Because each data point corresponds to a US state, the prediction can be explained by pointing to similar states with similar crime rates, providing an interpretability advantage over classical parametric symbolic regression.

### 5.6. Impact of Ensemble Selection

Figure 8 shows the effect of different ensemble selection strategies. Ensemble learning yields higher performance

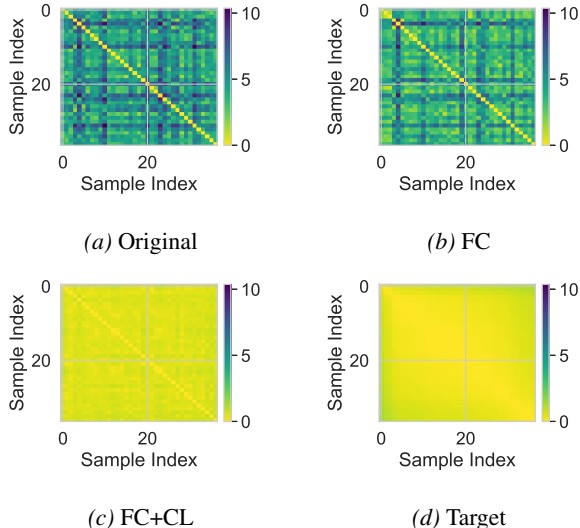

*(a)* Original      *(b)* FC

*(c)* FC+CL      *(d)* Target

*Figure 6.* Pairwise distance matrices across different feature spaces. FC denotes feature construction.

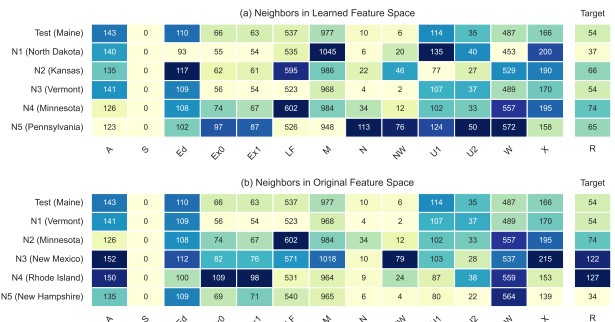

*Figure 7.* KNN retrieval on the US crime rate dataset. *Test* is the query instance; $N_1$–$N_5$ are its nearest neighbors from (a) the learned symbolic feature space and (b) the original feature space.

than single models, confirming its key role in stabilizing CSR. The improvement comes from variance reduction at two levels: on large datasets, subsampling introduces stochasticity in optimizing $W$, and ensemble averaging reduces this external variance; on small datasets, ensemble aggregation mitigates the internal variance of GP evolution. Even an ensemble of five individuals provides clear gains, showing that aggregation is critical for robustness.

The proposed DPP-based selection achieves a better balance between accuracy and diversity than top-$T$ selection or RBF-based DPP. DPP jointly considers model quality and semantic diversity through a kernel $L_{ij} = q_i q_j \mathrm{sim}(R_i, R_j)$, where $q_i$ is normalized fitness and $\mathrm{sim}$ is cosine similarity between residual semantics. This encourages diverse yet high-quality models and suppresses redundancy. The key advantage of DPP over top-$T$ selection is that top-$T$ selection greedily picks the highest-fitness individuals, which tend to be semantically similar and thus provide limited comple-

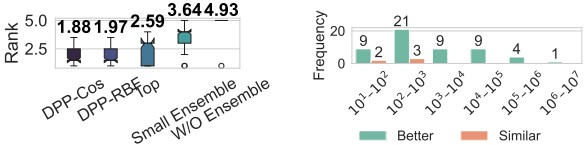

*Figure 8.* Average rank of meth- *Figure 9.* Effect of ensemble size ods based on test $R^2$ across on test $R^2$ across datasets. The datasets for different ensemble x-axis represents the number of variants.      instances for each dataset.

mentary information; DPP explicitly penalizes redundancy through the determinant-based kernel, ensuring that the selected ensemble covers diverse prediction behaviors while maintaining high quality.

Figure 9 summarizes the effect of ensemble size across datasets of different scales. Each bar represents the number of datasets where a larger ensemble achieves better or similar test $R^2$ than a single model. The results show that ensembles consistently improve performance on both small datasets, where $W$ is trained without subsampling, and large datasets, where stochastic optimization introduces variance. This confirms that DPP-based ensembles enhance stability and generalization regardless of dataset size.

## 6. Conclusions

This paper proposes Contrastive Symbolic Regression (CSR), a feature-construction-based symbolic regression framework integrating contrastive learning, adaptive KNN, and ensemble selection. First, contrastive learning with a closed-form solution transforms GP-constructed features into a target-aligned space where geometric distances reflect the structure of the target space, outperforming GP alone for KNN feature construction. Second, the adaptive KNN framework evaluates feature spaces through linear-rank weighted KNN, providing reliable fitness signals for evolutionary search, while efficient leave-one-out cross-validation selects the optimal $K$ without a large computational burden. Third, DPP-based ensemble selection preserves diverse top-performing models for robust performance. Experiments on 58 real-world regression datasets demonstrate that CSR consistently surpasses both modern symbolic regression and machine learning baselines. These findings show that combining symbolic feature construction with contrastive representation learning offers a promising direction for interpretable and effective regression modeling. A limitation is the focus on single-modality regression. Future work could extend to multi-modality and large-scale scenarios, and investigate alternative distance functions in CSR, although deriving closed-form solutions beyond Euclidean distance is non-trivial.

## Impact Statement

The proposed method is a general tabular learning method that can be applied to various domains. However, it is a predictive model, not a causal model, and therefore cannot imply that manipulating the identified variables would directly lead to the predicted results. The interpretable features constructed by our method provide an opportunity to scrutinize important features and understand their relationships with the target variable. For real-world usage, we suggest that practitioners not only focus on predictive accuracy but also carefully scrutinize the constructed features to avoid potential spurious correlations that may not generalize or may lead to incorrect causal interpretations.

The proposed method aligns with the goals of green artificial intelligence by producing compact symbolic representations that facilitate deployment on lower-cost embedded devices. From a green artificial intelligence perspective, the proposed method is a strong competitor to modern deep learning methods. It offers higher predictive accuracy, comparable training time, and a smaller model size.

## Acknowledgements

This work was supported in part by the Marsden Fund of New Zealand Government under Contract VUW1913, and Contract VUW2016; in part by the MBIE Data Science SSIF Fund under Contract RTVU1914; in part by the MBIE Endeavor Research Programme under Contract UOCX2104; and in part by the Catalyst: Leaders International Leader Fellowship grant under Contract 23-VUW-006-ILF.

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

**Overview.** The appendix supports the main paper as follows.

- Appendix A: Unified reference for mathematical symbols to support Section 3.
- Appendix B: Introduction to the full evolutionary pipeline to support Section 3.1.
- Appendix C: Methodological and runtime comparison with prior LOOCV schemes to complement Section 3.3.1.
- Appendix D: Pairwise significance tests for SRBench comparisons to complement Section 5.3.
- Appendix E: Illustrative interpretability example for constructed features to support Section 5.3.
- Appendix F: Ridge and contrastive importance definitions and fusion to support Appendix E.
- Appendix G: EQL-based initialization to support Section 3.1.
- Appendix H: GP and related hyperparameters to support Section 4.
- Appendix I: Sensitivity to L2 regularization weight to support Section 3.2.3.
- Appendix J: Vectorized formulation, closed-form solution, and recovery of $W$ to support Section 3.2.3.
- Appendix K: Laplacian regularization variant and rationale for contrastive learning alone to justify Section 3.2.3.
- Appendix L: Extended related work on tabular and foundation models to extend Section 2.
- Appendix M: Definitions of SR baselines and ablations to support Section 4.
- Appendix N: Robustness to subsample size when optimizing $W$ to support Section 5.2.
- Appendix O: Rationale for cosine kernel design and advantage of DPP-based over clustering-based ensemble selection to support Section 3.4.
- Appendix P: Rationale for linear-rank weighted KNN to justify Section 3.3.2.
- Appendix Q: Tuning protocol for baseline methods to support Section 4.

## A. Notation

Table 3 summarizes the mathematical notation used throughout this paper.

## B. Algorithm Framework

The proposed algorithm is built upon an evolutionary feature construction framework. As illustrated in Figure 2, it consists of five stages:

- **Random Initialization:** A population of $P$ candidate solutions is initialized, each consisting of $m$ symbolic trees representing an $m$-dimensional feature space. The initialization combines a sparse neural network–based symbolic regression model with a random initialization method from the genetic programming literature. This combination leverages the efficient sparse neural network to quickly obtain a good starting point for evolutionary search, while the random initialization ensures population diversity:
  - **EQL-based initialization:** A sparse neural network, the Equation Learner Network (EQL) (Zhang et al., 2023c), is used to generate one individual. The details are provided in Appendix G.
  - **Random initialization:** The remaining individuals are generated by the ramped half-and-half procedure (Banzhaf et al., 1998), which balances depth-limited and fully expanded trees to maintain structural diversity.
- **Solution Evaluation:** Using the $m$ GP trees, $m$ features are constructed from the original data $X$. A ridge regression model is first fitted on these constructed features to capture the global linear structure. The features are then transformed via a learned transformation matrix $W$, and KNN regression is applied to the transformed features to model the residuals, as described in the contrastive symbolic regression model introduced in Section 3.2. The $R^2$ on the training data is used as the fitness value.

*Table 3.* Summary of mathematical notation.

| Symbol | Description |
| --- | --- |
| *Data and Features* | |
| $X \in \mathbb{R}^{n \times d}, x_i$ | Input feature matrix and $i$-th instance |
| $y \in \mathbb{R}^n, y_i, \hat{y}_i$ | Target vector, value, and prediction |
| $n, d, m$ | Number of samples, original features, constructed features |
| $\Phi(X), \phi_j(X)$ | Constructed feature matrix and $j$-th feature |
| *Contrastive Learning* | |
| $W \in \mathbb{R}^{m \times m}$ | Transformation matrix |
| $z_i, Z$ | Transformed feature for instance $i$ and matrix |
| $D, D'$ | Target-based and feature-based distance matrices |
| *KNN and Regression* | |
| $K, K^\star$ | Number of neighbors and optimal $K$ |
| $\mathcal{N}_K(z_i)$ | Set of $K$-nearest neighbors |
| $\beta, e, \lambda$ | Ridge coefficients, ridge regression residuals, regularization |
| *Evolution and Ensemble* | |
| $P, p_i$ | Population and $i$-th individual |
| $\mathcal{C}$ | Candidate set for DPP-based ensemble selection |
| $A, L$ | Archive and DPP kernel matrix |
| $q_i, R_i$ | Normalized fitness and residual semantics |
| $S$ | Subsample size used to optimize $W$ |
| $E$ | Ensemble size |
| $\mathcal{S}$ | Set of selected individuals in DPP |

- **Parent Selection:** Promising individuals are selected using double lexicase selection (Zhang et al., 2023a). The key idea of double lexicase selection is to apply lexicase selection twice to choose one individual. Each lexicase selection (La Cava et al., 2019a) filters candidates based on per-instance performance criteria. The selection criterion is defined as $\epsilon + \text{MAE}$, where MAE denotes the median absolute error. In the second round, the two candidates are selected based on the inverse of tree size to encourage the choice of parsimonious individuals for better interpretability.

- **Offspring Generation:** New GP trees are generated through random subtree crossover and mutation. Each individual contains $m$ trees, and both operations are performed $m$ times according to their respective probabilities to encourage exploration.

- **Ensemble Selection:** At the end of each generation, the top-performing individuals from the current population are merged with the archive, and DPP-based selection preserves diverse high-quality individuals, as described in Section 3.4.

## C. Difference from Existing LOOCV-KNN and Runtime Analysis

**Methodological Difference.** All methods in this study use FAISS-based nearest-neighbor search with a flat (L2) index, where each query has complexity $\mathcal{O}(n)$. The key difference lies in how they reuse neighbor information and how many models they must fit. The Naive LOOCV explicitly retrains the KNN regressor once per sample for each candidate $k$, resulting in $n \times K$ model fits. The existing TMLR-LOOCV method achieves exact leave-one-out evaluation by leveraging predictions from $(k+1)$-NN models and applying a correction (Kanagawa, 2024), requiring only $K$ model fits. Specifically, it computes the leave-one-out mean squared error (MSE) as

$$\widehat{\text{MSE}}_k^{\text{TMLR-LOOCV}} = \frac{1}{n} \sum_{i=1}^n (y_i - \hat{y}_i^{(k+1)})^2 \cdot \frac{(k+1)^2}{k^2}, \tag{10}$$

where $\hat{y}_i^{(k+1)}$ denotes the $(k+1)$-NN prediction for sample $i$. In contrast, our proposed method computes all candidate $k$ values simultaneously in a single vectorized pass. Given all neighbor information $\{y_{i,j}\}$, cumulative predictions are computed as $\hat{y}_i^{(k)} = \frac{1}{k} \sum_{j=1}^k y_{i,j}$ for $k = 1, \ldots, K_{\max}$, and the corresponding mean squared errors are $\text{MSE}_k = \frac{1}{n} \sum_{i=1}^n (y_i - \hat{y}_i^{(k)})^2$. The optimal $K^\star$ is selected by maximizing the average leave-one-out $R^2$. This single-pass design preserves exact LOOCV

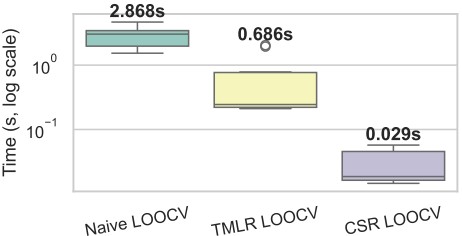

*Figure 10.* Runtime comparison (log scale) among Naive LOOCV, TMLR-LOOCV, and CSR LOOCV over ten runs.

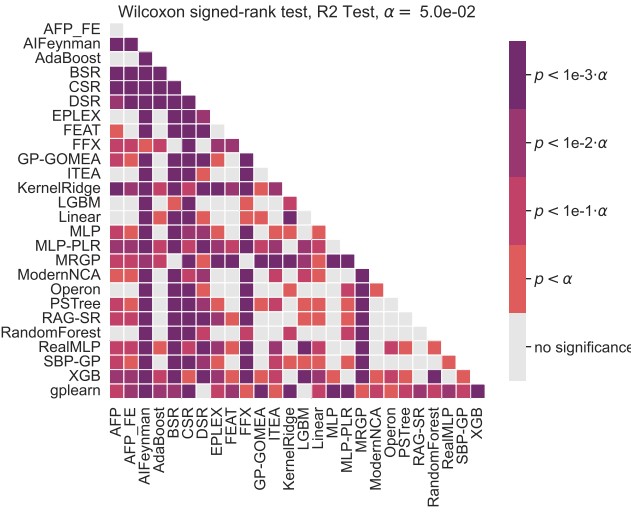

*Figure 11.* Pairwise statistical comparison of test $R^2$ scores over 58 black-box problems.

while reducing the number of required model fits from $n \times K$ (Naive) and $K$ (TMLR-LOOCV) to just one, achieving low computational overhead.

**Runtime Analysis.** Figure 10 compares the runtime of Naive LOOCV, TMLR-LOOCV, and the proposed CSR LOOCV over ten independent runs using 100 samples and 10 features. The Naive LOOCV is the slowest by several orders of magnitude because it refits $n$ models for each $k$. The TMLR-LOOCV substantially improves efficiency by requiring only $K$ model fits through reuse of $(k+1)$-NN predictions. The proposed CSR LOOCV achieves the lowest runtime by fitting only once and evaluating all $k$ values through shared neighbor information. These results confirm that our method maintains exact LOOCV while achieving low computational overhead.

## D. Pairwise Statistical Comparison of Test $R^2$

Figure 4 shows that CSR achieves superior overall performance. In this section, we further perform a Wilcoxon signed-rank test with Benjamini–Hochberg correction to verify the statistical significance of the proposed method over other methods. The results, shown in Figure 11, indicate that the superiority of CSR over the other algorithms is statistically significant.

## E. Feature Importance Analysis

The ensemble model in CSR is not directly visualizable because predictions are produced by multiple models rather than a single explicit equation. Nevertheless, the importance of constructed features can be analyzed following the methods described in Appendix F. This form of feature-level explainability is also used in feature transformation methods such as FastFT (He et al., 2025), which likewise analyze importance scores assigned to constructed features. With all original features normalized, an illustrative example is shown in Figure 12 using the interaction feature $\min(X_3, \max(X_{10}, X_9))$, where $X_3$ represents the mean years of schooling, $X_{10}$ denotes the unemployment rate of young urban males, and $X_9$ reflects

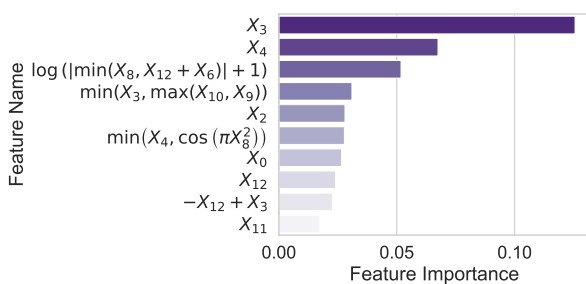

Figure 12. Visualization of feature importance in the constructed feature space.

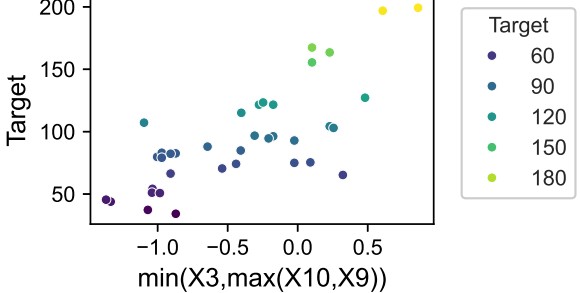

Figure 13. Relationship between the constructed feature and crime rate.

the non-white population per 1000. Interestingly, this feature exhibits a positive correlation with crime rates, suggesting that higher education levels—in the context of high-risk socioeconomic factors like youth unemployment ($X_{10}$) or racial density ($X_9$)—can act as proxies for urbanization or greater potential rewards for criminal activity. This conditional pattern is captured by the interaction term $\min(X_3, \max(X_{10}, X_9))$, providing discriminative signals that would be difficult to identify using individual features alone. By encoding these relationships, the interaction feature enhances the predictive accuracy of KNN and allows domain experts to interpret learned representations. Crucially, CSR is a predictive model rather than a causal one; while these interactions can help guide resource allocation, such as identifying areas where law enforcement presence may need to be strengthened, they do not imply that manipulating these variables would directly reduce crime rates.

## F. Feature Importance in CSR

We analyze feature importance in the CSR model, which learns a ridge regression component jointly with a contrastive learning module. Both stages learn on the constructed symbolic feature space $\Phi(X)$, where each feature is standardized by z-score normalization to make the learned weights reflect relative importance across dimensions. The resulting importance captures both linear predictive strength and geometric influence within the learned contrastive space.

**Ridge-based importance.** The ridge component fits a linear mapping $y = \Phi(X)\beta$, where $\beta \in \mathbb{R}^m$ are regression coefficients. The importance of feature $i$ is computed from its absolute coefficient magnitude $I_i^{(\text{ridge})} = |\beta_i|$, normalized as $\tilde{I}_i^{(\text{ridge})} = |\beta_i| / \sum_j |\beta_j|$. This measures the direct linear contribution of each constructed feature to the predictive output.

**Contrastive importance.** The residuals $e = y - \hat{y}_{\text{ridge}}$ are modeled through a contrastive learning module that learns a transformation matrix $W \in \mathbb{R}^{m \times m}$, mapping $\Phi(X)$ to a latent space $Z = \Phi(X)W$ where distances reflect target similarities. The contribution of feature $i$ is quantified by its squared $\ell_2$-norm across latent dimensions, $I_i^{(\text{ctr})} = \|W_{i,\cdot}\|_2^2$, and normalized as $\tilde{I}_i^{(\text{ctr})} = I_i^{(\text{ctr})} / \sum_j I_j^{(\text{ctr})}$. This term captures how each feature shapes the contrastive embedding geometry.

**Adaptive fusion.** The two importance measures are combined using a data-driven adaptive weight

$$\lambda = \frac{\text{std}(|\beta|)}{\text{std}(|\beta|) + \text{std}([\|W_{1,\cdot}\|_2^2, \ldots, \|W_{m,\cdot}\|_2^2])}, \tag{11}$$

and the final per-feature importance is defined as

$$I_i = \lambda \, \tilde{I}_i^{(\text{ridge})} + (1 - \lambda) \, \tilde{I}_i^{(\text{ctr})}. \tag{12}$$

The factor $\lambda$ automatically balances the influence of the ridge and contrastive components: when the ridge coefficients exhibit higher variability across features, the predictive term dominates; otherwise, the contrastive term contributes more.

**Ensemble aggregation.** When multiple CSR learners are trained on different subsets of constructed features, the ensemble-level importance is obtained by averaging across all $E$ members,

$$\bar{I}_i = \frac{1}{E} \sum_{r=1}^{E} I_i^{(r)}. \tag{13}$$

This provides a measure of feature importance for the ensemble.

## G. Equation Learner for Initialization

**Model formulation.** The deep Equation Learner (EQL) (Zhang et al., 2023c) is a symbolic regression method based on neural networks. Since gradient-based optimization makes it difficult to perform true $\ell_0$ optimization, the EQL employs $\ell_{0.5}$ regularization to encourage sparsity. Although EQL does not achieve state-of-the-art performance compared to search-based methods such as evolutionary algorithms or Monte Carlo tree search, it is computationally efficient and can serve as a fast initialization method for evolutionary algorithms. Specifically, the EQL minimizes a sparsity-regularized mean squared error loss to encourage compact symbolic representations:

$$\mathcal{L}_{\text{EQL}}(\theta) = \frac{1}{n} \sum_{i=1}^{n} \left( f_\theta(x_i) - y_i \right)^2 + \lambda \sum_j \left| \theta_j \right|^{0.5}, \tag{14}$$

where $f_\theta$ denotes the neural network output, $\theta_j$ are the trainable parameters, and $\lambda$ is the regularization weight. The fractional $\ell_{0.5}$ penalty serves as a continuous relaxation of $\ell_0$ regularization, promoting sparsity in the learned weights while maintaining differentiability for gradient-based optimization.

**Initialization strategy.** In this paper, we use EQL to generate seed solutions for GP. Specifically, since each individual consists of 10 GP trees, we employ a bootstrapping approach to train 10 EQL models on 10 bootstrapped datasets, and then convert the learned EQL models into GP trees to form the 10 trees of an individual. The intuition is that these 10 EQL models capture different aspects of the underlying data-generating process, and subsequent base learners such as linear regression and KNN can leverage these diverse representations to improve predictive performance. For datasets with more than five features, features are randomly subsampled to five per round, as EQL tends to struggle to produce sparse solutions when the feature dimensionality is high. The training settings of EQL are presented in Table 4.

| Parameter | Value |
|---|---|
| Layers | 1 |
| Regularization Weight | $5 \times 10^{-3}$ |
| Learning Rate | $1 \times 10^{-2}$ |
| Epochs | 20000 |
| Batch Size | 16 |
| Validation Split | 0.2 |
| Patience | 20 |
| Minimum Delta | $1 \times 10^{-6}$ |

*Table 4.* Key Hyperparameters for the Equation Learner Networks.

## H. Parameter Settings of Genetic Programming

The hyperparameters of the GP-based framework largely follow standard practice, as summarized in Table 5. Each individual consists of ten trees, with population size set to 200 and the number of generations set to 50. Crossover and mutation probabilities are fixed at 0.9 and 0.1, respectively. The initial tree depth is randomly initialized between 0 and 2 and capped at 10 to prevent bloat. To ensure numerical stability, protected operators are used: division is defined as the analytical quotient $AQ(a, b) = a/\sqrt{1 + b^2}$ (Ni et al., 2012), logarithm as $\log(|x| + 1)$, and square root as $\sqrt{|x|}$. For linear-rank weighted KNN during feature evaluation, the number of neighbors is set to $K = 5$ for datasets with 200 or fewer samples and $K = 30$ for larger datasets, using smaller neighborhoods when data are limited and larger neighborhoods when more samples are available. Final prediction uses the LOOCV-selected $K^\star$. The ensemble size is fixed to 30.

## I. Analysis on $\ell_2$ Regularization Weights

Figure 14 shows the impact of different $\ell_2$ regularization weights $\lambda$. The results indicate that while tuning the regularization weight $\lambda$ can lead to minor variations across datasets, the differences are generally small, suggesting that the method is not highly sensitive to the choice of $\ell_2$ regularization.

*Table 5.* Parameter settings for GP-based feature construction.

| Parameter | Value |
|---|---|
| Population size | 200 |
| Number of generations | 50 |
| Crossover / Mutation rate | 0.9 / 0.1 |
| Initial tree depth | 0–2 |
| Maximum tree depth | 10 |
| Number of trees per individual | 10 |
| Contrastive regularization $\lambda$ | 0.1 |
| Ensemble size | 30 |

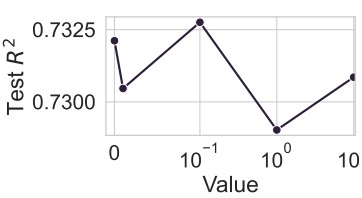 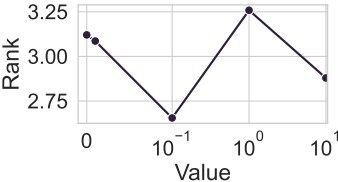

*Figure 14.* Effect of $\ell_2$ regularization with different weights. Both plots show aggregated results over 58 datasets with median of 30 runs. The left plot shows mean test $R^2$ scores, the larger the better. The right plot shows mean rank, the smaller the better.

## J. Optimization of the Transformation Matrix and Closed-form Contrastive Solution

The objective is to learn a transformation matrix $W \in \mathbb{R}^{m \times m}$ such that the transformed distance matrix $D' \in \mathbb{R}^{n \times n}$ approximates the target distance matrix $D \in \mathbb{R}^{n \times n}$. Expanding the squared distance yields

$$
\begin{aligned}
D'_{ij} &= \|W^\top \Phi(x_i) - W^\top \Phi(x_j)\|^2 \\
&= (\Phi(x_i) - \Phi(x_j))^\top WW^\top (\Phi(x_i) - \Phi(x_j)),
\end{aligned}
\tag{15}
$$

where $\Phi(x_i) \in \mathbb{R}^m$ denotes the GP-constructed feature vector for sample $i$.

### J.1. Vectorized Least-Squares Formulation

Introducing $M = WW^\top \in \mathbb{R}^{m \times m}$, the pairwise distance term can be expressed as

$$
D'_{ij} = (\Phi(x_i) - \Phi(x_j))^\top M (\Phi(x_i) - \Phi(x_j)).
\tag{16}
$$

For notational convenience, define

$$
b_{i,j} = \text{vec}\big((\Phi(x_i) - \Phi(x_j))(\Phi(x_i) - \Phi(x_j))^\top\big) \in \mathbb{R}^{m^2},
\tag{17}
$$

and $\mathbf{v} = \text{vec}(M) \in \mathbb{R}^{m^2}$. Then, each pairwise distance can be written as a linear function:

$$
D'_{ij} = \mathbf{v}^\top b_{i,j}.
\tag{18}
$$

Stacking all pairwise terms yields the regularized least-squares objective:

$$
\mathcal{L}(\mathbf{v}) = \|d - B\mathbf{v}\|^2 + \lambda \|\mathbf{v}\|^2,
\tag{19}
$$

where $d = \text{vec}(D) \in \mathbb{R}^{n^2}$ is the vectorized target distance matrix and $B \in \mathbb{R}^{n^2 \times m^2}$ encodes all pairwise feature differences.

### J.2. Detailed Gradient Derivation and Closed-form Solution

Expanding the loss in Equation (19) gives

$$
\begin{aligned}
\mathcal{L}(\mathbf{v}) &= (d - B\mathbf{v})^\top (d - B\mathbf{v}) + \lambda \mathbf{v}^\top \mathbf{v} \\
&= d^\top d - 2d^\top B\mathbf{v} + \mathbf{v}^\top B^\top B\mathbf{v} + \lambda \mathbf{v}^\top \mathbf{v}.
\end{aligned}
\tag{20}
$$

Taking the gradient of $\mathcal{L}(\mathbf{v})$ with respect to $\mathbf{v}$ yields

$$\nabla_{\mathbf{v}}\mathcal{L}(\mathbf{v}) = -2B^{\top}d + 2B^{\top}B\mathbf{v} + 2\lambda\mathbf{v}. \tag{21}$$

Setting the gradient to zero for optimality:

$$\nabla_{\mathbf{v}}\mathcal{L}(\mathbf{v}) = 0 \quad \Rightarrow \quad (B^{\top}B + \lambda I)\mathbf{v} = B^{\top}d. \tag{22}$$

Solving this linear system gives the closed-form ridge-regularized solution:

$$\mathbf{v}^* = (B^{\top}B + \lambda I)^{-1}B^{\top}d, \quad \mathbf{v}^* \in \mathbb{R}^{m^2}. \tag{23}$$

This expression provides a stable and unique estimator for the vectorized transformation parameters without iterative optimization.

### J.3. Recovering the Transformation

The optimal vector $\mathbf{v}^*$ is reshaped into $M^* = \mathrm{unvec}(\mathbf{v}^*)$ and symmetrized for numerical stability. We then obtain $W$ via eigendecomposition:

$$M^* = U\Lambda U^{\top}, \quad W = U\Lambda_+^{1/2}, \tag{24}$$

where $U$ contains the eigenvectors and $\Lambda$ is diagonal. We form $\Lambda_+$ by clamping negative eigenvalues in $\Lambda$ to zero before taking the square root. The resulting $W$ provides an efficient linear mapping that aligns the transformed feature distances with the target-based distance structure.

## K. Graph Laplacian Regularization

Graph Laplacian regularization is a potential approach to further improve the learned feature space by preserving local neighborhood structures. In this section, we explore combining Graph Laplacian regularization with contrastive learning to investigate whether this additional regularization can enhance performance or whether contrastive learning alone is sufficient.

### K.1. Formulation

In nearest-neighbor methods, predictive performance depends on preserving the relative ranking of distances rather than their absolute magnitude. To encourage instances with similar targets to remain close after transformation, we introduce a Laplacian regularization term that preserves the neighborhood structure implied by target similarities.

A target-similarity matrix is constructed using a Gaussian kernel:

$$S_{ij} = \exp\left(-\frac{(y_i - y_j)^2}{2\sigma^2}\right), \tag{25}$$

where $\sigma$ controls how rapidly similarity decays with increasing target difference. The corresponding degree matrix is defined as $\mathcal{D}_{ii} = \sum_j S_{ij}$, and the graph Laplacian is $L = \mathcal{D} - S$, where $S, \mathcal{D}, L \in \mathbb{R}^{n \times n}$.

The Laplacian regularization term is defined as

$$\mathcal{L}_{\mathrm{lap}} = \sum_{i,j} S_{ij} \|W^{\top}\Phi(x_i) - W^{\top}\Phi(x_j)\|^2, \tag{26}$$

where $\Phi(x_i) \in \mathbb{R}^m$ is the feature vector for sample $i$, and $W \in \mathbb{R}^{m \times m}$ is the transformation matrix to be learned. This term encourages samples with large $S_{ij}$, corresponding to similar targets, to stay close in the transformed feature space $\Phi(X)W \in \mathbb{R}^{n \times m}$, thereby preserving target-induced neighborhoods and improving KNN consistency.

Combining this term with the contrastive alignment loss yields the overall objective:

$$\min_{M} \|D' - D\|^2 + \lambda_1\|\mathrm{vec}(M)\|^2 + \lambda_G\mathcal{L}_{\mathrm{lap}}, \quad M = WW^{\top}, \tag{27}$$

where $D', D \in \mathbb{R}^{n \times n}$ are the transformed and target distance matrices.

## K.2. Closed-form Solution with Laplacian Regularization

Let $\Delta_{ij} = \Phi(x_i) - \Phi(x_j) \in \mathbb{R}^m$ denote the feature difference between two samples. The squared distance between their transformed representations expands as

$$\|W^\top \Phi(x_i) - W^\top \Phi(x_j)\|^2 = \Delta_{ij}^\top W W^\top \Delta_{ij}. \tag{28}$$

Define $M = WW^\top \in \mathbb{R}^{m \times m}$ as a symmetric matrix. Substituting into Equation (26) gives

$$\mathcal{L}_{\text{lap}} = \sum_{i,j} S_{ij} \, \Delta_{ij} M \Delta_{ij}^\top. \tag{29}$$

**Expanding the Laplacian Form.**   Let $\Phi(X) = [\phi_1^\top, \ldots, \phi_n^\top]^\top \in \mathbb{R}^{n \times m}$ with $\phi_i = \Phi(x_i)$. The Laplacian matrix $L = \mathcal{D} - S$ captures how each sample connects to target-similar neighbors. We can expand the matrix product $\Phi(X)^\top L \Phi(X)$ step by step:

$$\Phi(X)^\top \mathcal{D} \Phi(X) = \sum_{i=1}^{n} \mathcal{D}_{ii} \, \phi_i^\top \phi_i = \sum_{i,j} S_{ij} \, \phi_i^\top \phi_i, \tag{30}$$

$$\Phi(X)^\top S \Phi(X) = \sum_{i,j} S_{ij} \, \phi_i^\top \phi_j. \tag{31}$$

Subtracting these gives

$$\Phi(X)^\top L \Phi(X) = \sum_{i,j} S_{ij} \, (\phi_i^\top \phi_i - \phi_i^\top \phi_j). \tag{32}$$

Using the symmetry of $S$ and adding the same term with $(i,j)$ swapped yields the familiar pairwise-difference form:

$$\Phi(X)^\top L \Phi(X) = \tfrac{1}{2} \sum_{i,j} S_{ij} \, (\phi_i - \phi_j)^\top (\phi_i - \phi_j) = \tfrac{1}{2} \sum_{i,j} S_{ij} \, \Delta_{ij}^\top \Delta_{ij}. \tag{33}$$

The constant factor $\tfrac{1}{2}$ can be absorbed into $\lambda_G$ without affecting the optimization.

**Vectorized Representation.**   Using the compact form $\Phi(X)^\top L \Phi(X) = \sum_{i,j} S_{ij} \Delta_{ij}^\top \Delta_{ij}$, the Laplacian regularization can be expressed as

$$\mathcal{L}_{\text{lap}} = \text{Tr}\big(M \Phi(X)^\top L \Phi(X)\big), \qquad M = WW^\top \in \mathbb{R}^{m \times m}. \tag{34}$$

Applying the identity $\text{Tr}(A^\top B) = \text{vec}(A)^\top \text{vec}(B)$ and defining $\mathbf{v} = \text{vec}(M) \in \mathbb{R}^{m^2}$, we obtain

$$\mathcal{L}_{\text{lap}} = g^\top \mathbf{v}, \qquad g = \text{vec}\big(\Phi(X)^\top L \Phi(X)\big) \in \mathbb{R}^{m^2}. \tag{35}$$

Substituting this term into the contrastive objective yields the vectorized quadratic form:

$$\mathcal{L}(\mathbf{v}) = (B\mathbf{v} - d)^\top (B\mathbf{v} - d) + \lambda_1 \mathbf{v}^\top \mathbf{v} + \lambda_G g^\top \mathbf{v}, \tag{36}$$

where $B \in \mathbb{R}^{n^2 \times m^2}$ encodes all pairwise feature interactions, and $d = \text{vec}(D) \in \mathbb{R}^{n^2}$ is the vectorized target distance matrix.

Taking the gradient with respect to $\mathbf{v}$ and setting it to zero gives

$$\nabla_{\mathbf{v}} \mathcal{L}(\mathbf{v}) = 2B^\top (B\mathbf{v} - d) + 2\lambda_1 \mathbf{v} + \lambda_G g = 0, \tag{37}$$

which leads to the normal equation

$$(B^\top B + \lambda_1 I_{m^2})\mathbf{v} = B^\top d - \tfrac{\lambda_G}{2} g. \tag{38}$$

The closed-form solution is therefore

$$\mathbf{v}^* = (B^\top B + \lambda_1 I_{m^2})^{-1} \big(B^\top d - \tfrac{\lambda_G}{2} g\big), \tag{39}$$

where $\mathbf{v}^* \in \mathbb{R}^{m^2}$ represents the optimal vectorized metric transformation. Reshaping $\mathbf{v}^*$ into $M^* = \text{unvec}(\mathbf{v}^*) \in \mathbb{R}^{m \times m}$, symmetrizing, and decomposing as $M^* = U\Lambda U^\top$ yields the final transformation $W = U\Lambda_+^{1/2} \in \mathbb{R}^{m \times m}$, which enforces target-consistent neighborhood smoothness in the learned space.

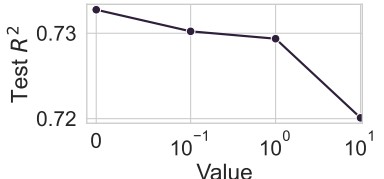 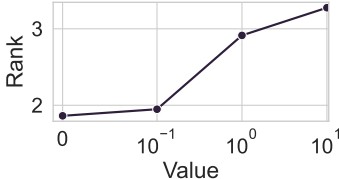

*Figure 15.* Effect of Laplacian regularization with different weights. Both plots show aggregated results over 58 datasets with median of 30 runs. The left plot shows mean test $R^2$ scores, the larger the better. The right plot shows mean rank, the smaller the better.

### K.3. Experimental Results

In this section, we investigate the effect of Laplacian regularization with different weights $\lambda_G \in \{0, 0.1, 1, 10\}$, where $\lambda_G = 0$ corresponds to no regularization. Figure 15 shows that model performance gradually decreases as $\lambda_G$ increases, indicating that Laplacian regularization is not beneficial for CSR and may even degrade performance. A key limitation is that the Laplacian term $\mathcal{L}_{\text{lap}} = \sum_{i,j} S_{ij} \|W^\top \Phi(x_i) - W^\top \Phi(x_j)\|^2$ only pulls neighboring samples together without pushing dissimilar ones apart. When $\lambda_G$ becomes large, this attraction dominates, and all transformed representations $W^\top \Phi(x_i)$ might collapse to a constant vector, oversmoothing the constructed feature space and reducing its discriminative capacity. In Locality Preserving Projection (He & Niyogi, 2003), the constraint $W^\top \Phi(X_p)^\top \mathcal{D} \Phi(X_p) W = I$ is applied to prevent such collapse, but integrating this constraint into the joint contrastive and Laplacian formulation while retaining a closed-form solution is non-trivial. Overall, the empirical results suggest that using contrastive learning alone, without Laplacian regularization, is a more effective choice for contrastive symbolic regression.

## L. Related Work on Tabular Data Learning

Tabular data learning has witnessed remarkable advances in recent years, driven by the development of deep architectures and pretraining strategies. Early methods sought to model heterogeneous features effectively: TabNet (Arik & Pfister, 2021) introduced attentive feature selection for interpretability, while FT-Transformer (Gorishniy et al., 2021), SAINT (Somepalli et al., 2022), and NODE (Popov et al., 2020) leveraged attention and differentiable decision ensembles to capture cross-feature interactions. Follow-up work emphasized model simplicity and strong regularization, as RTDL (Gorishniy et al., 2021) and RealMLP (Holzmüller et al., 2024) showed that optimized MLPs can rival complex transformers. Other studies advanced explicit feature interaction modeling: AMFormer (Cheng et al., 2024) augments transformers with additive and multiplicative attention to model arithmetic feature interactions, BiSHop (Xu et al., 2024a) applies sparse, Entmax-based attention derived from Hopfield dynamics for tabular learning, ExcelFormer (Chen et al., 2024a) introduces structured gating for numerical stability, and GRANDE (Marton et al., 2024) together with DOFEN (Chen et al., 2024b) unify neural and decision tree-based ensemble paradigms. To improve transferability, cross-table pretraining methods such as TransTab (Wang & Sun, 2022) and XTab (Zhu et al., 2023) learn table-invariant representations via contrastive and reconstructive objectives, and CM2 (Ye et al., 2024) further optimizes reconstruction loss based on cosine similarity to align heterogeneous tables. The integration of language models has also enabled semantic understanding of tabular features. TP-BERTa (Yan et al., 2024) adapts RoBERTa to tabular learning by converting numerical features into discrete tokens, CM2 (Ye et al., 2024) learns column-name embeddings concatenated with value embeddings to capture semantic information of each column, CARTE (Kim et al., 2024) uses graph neural networks to learn general representations across tables based on FastText-encoded features, and FeatLLM (Han et al., 2024) employs language models to construct binary features that enhance the performance of linear classifiers. Finally, foundation-style models including TabPFN (Hollmann et al., 2025), UniTabE (Yang et al., 2024), TabuLa-8B (Gardner et al., 2024), TabM (Gorishniy et al., 2025), HyperFast (Bonet et al., 2024), and MotherNet (Mueller et al., 2025) scale tabular modeling through parameter-efficient pretraining, ensembling, and in-context reasoning. Despite these advances, symbolic regression remains a desirable algorithm due to its interpretability, inference efficiency, and low memory requirements. Symbolic regression methods explicitly construct interaction features using mathematical operations, offering transparent representations that help experts interpret predictions. Moreover, the number of symbolic features is typically orders of magnitude smaller than the weight matrices of deep neural networks, resulting in faster inference and reduced memory consumption—an appealing property for deployment on embedded devices.

# M. Symbolic Regression Variants

This section provides detailed descriptions of the symbolic regression variants introduced in Section 4:

- **SR$^{\text{LR}}$**: Linear regression is used as the base learner, consistent with ITEA (de França & Aldeia, 2021), M3GP (Muñoz et al., 2019), FEAT (La Cava & Moore, 2020), and FEAT-KD (Fong & Motani, 2025). Ridge regression is adopted to better control overfitting.

- **SR$^{\text{KNN}}$**: Direct feature construction using a naive KNN regressor, corresponding to M4GP (La Cava et al., 2019b).

- **SR$^{\text{CL+KNN}}$**: A variant using contrastive learning transformation with KNN, without ridge regression. This demonstrates the benefit of contrastive learning for feature space alignment in pure KNN.

- **SR$^{\text{DT}}$**: Direct feature construction using a decision tree regressor, following Evolutionary Forest (Zhang et al., 2021).

- **SR$^{\text{LR+KNN}}$**: A hybrid variant where ridge regression first captures global linear structure, and KNN subsequently models residuals. Serves as an ablation baseline without contrastive learning.

- **SR$^{\text{LR+DT}}$**: A hybrid variant where ridge regression captures global linear structure, and the decision tree models residuals (Zhang et al., 2023b).

- **SR$^{\text{LR+PLS+KNN}}$**: A variant using Partial Least Squares (PLS) (Wold et al., 1984) to optimize a linear transformation that maximizes covariance between transformed features and targets:

$$\arg\max_{w} \frac{(w^\top \Phi(X)^\top y)^2}{w^\top \Phi(X)^\top \Phi(X)w}. \tag{40}$$

  Multiple PLS components are extracted sequentially with deflation and combined to form a transformation matrix $W \in \mathbb{R}^{m \times m}$, as in DeKPLS (Chen & Wang, 2024). KNN is then applied in this transformed space to assess the predictive performance of PLS relative to contrastive learning.

- **SR$^{\text{LR+LPP+KNN}}$**: A variant using Locality Preserving Projection (LPP) (He & Niyogi, 2003) to preserve local neighborhood relationships. LPP is applied to a subset of $S$ randomly chosen instances. Given an adjacency matrix $A \in \mathbb{R}^{S \times S}$ and Laplacian $L = \mathcal{D} - A$ with $\mathcal{D}_{ii} = \sum_j A_{ij}$,

$$\arg\min_{W} \sum_{i,j=1}^{S} A_{ij} \|W^\top \Phi(x_i) - W^\top \Phi(x_j)\|^2, \tag{41}$$

  equivalently expressed as $\arg\min_W \text{Tr}(W^\top \Phi(X_S)^\top L \Phi(X_S)W)$, subject to $W^\top \Phi(X_S)^\top \mathcal{D}\Phi(X_S)W = I$. This ensures nearby samples remain close after transformation. KNN is then applied in the transformed space to compare LPP-based and contrastive mappings.

- **SR$^{\text{LR+CL+KNN}}$**: A contrastive learning variant where the transformation matrix is optimized via contrastive loss. Ridge regression captures global linear trends, and KNN models residuals in the contrastively transformed feature space.

- **SR$^{\text{LOO-KNN}}$**: A variant using KNN with leave-one-out cross-validation for adaptive $K$ selection, without contrastive learning transformation.

- **SR$^{\text{CL+LOO-KNN}}$**: A variant combining contrastive learning transformation with KNN using leave-one-out cross-validation for adaptive $K$ selection. This variant demonstrates the benefit of contrastive learning with adaptive KNN.

- **SR$^{\text{All}}$**: The full variant combining ridge regression, contrastive transformation, and linear-rank weighted KNN with efficient LOOCV-based $K$ selection.

*Table 6.* Statistical comparison of **test $R^2$ scores** of GP-based models with different subsample sizes. Each comparison reports the number of datasets where the row subsample size is significantly better, statistically tied, or significantly worse than the column subsample size.

|      | 150 | 200 |
|------|-----|-----|
| 100  | 3(+)/44($\sim$)/11(-) | 1(+)/40($\sim$)/17(-) |
| 150  | — | 0(+)/53($\sim$)/5(-) |

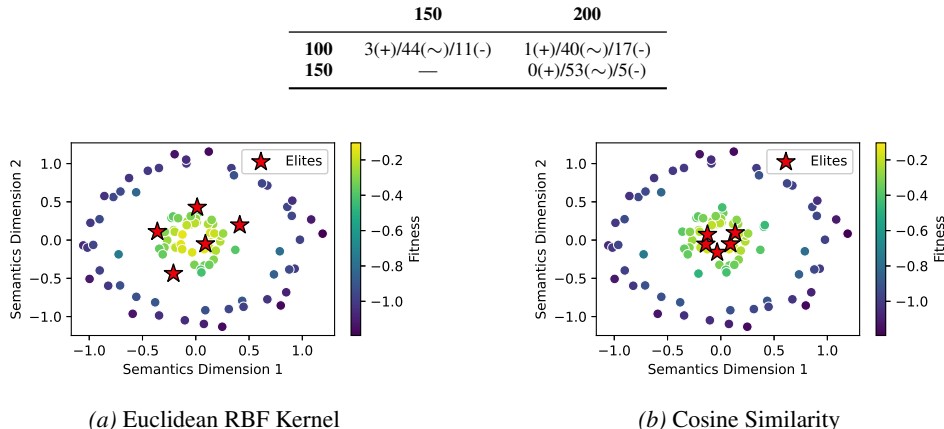

*(a)* Euclidean RBF Kernel        *(b)* Cosine Similarity

*Figure 16.* DPP-based ensemble selection under Euclidean and cosine kernels. Each point represents an individual in the semantics/output space.

## N. Impact of Subsampling Strategy

The subsampling strategy directly affects the optimization of $W$. To evaluate its influence, we compare subsample sizes of 100, 150, and 200 instances. The differences are statistically insignificant on most datasets, with many ties, indicating that CSR is relatively robust to this choice. Nevertheless, increasing the subsample size can further improve performance: 150 and 200 tend to outperform 100, and 200 also tends to outperform 150. In our experiments, we use 100 as the default subsample size, which already provides strong performance across a large number of datasets while keeping the optimization efficient; if computational resources permit, a larger subsample such as 200 is preferable.

## O. Kernel Design and Advantage of DPP-Based Ensemble Selection

### O.1. Effect of Distance Measure on DPP Selection

Figure 16a and Figure 16b illustrate a potential scenario of how the choice of distance measure affects DPP-based ensemble selection. Each dot represents an individual, and its position represents the predicted output on two instances. Fitness is calculated as the squared distance to the origin, which represents the target. The choice of kernel affects which individuals are selected, as it determines how diversity is measured among candidates. As shown, the cosine similarity kernel selects a more concentrated set of high-fitness individuals, while the Euclidean RBF kernel introduces spurious diversity by including low-fitness outliers far from the origin.

### O.2. Limitation of Clustering-Based Ensemble Selection

A straightforward idea for ensemble selection is to apply quality–diversity optimization methods such as CVT-MAP-Elites (Vassiliades et al., 2017). The basic idea is a two-stage selection process: first, clustering partitions the semantic space into regions, and then the best-performing individual is selected from each cluster. This idea has been instantiated in diversity-based selective ensemble methods (Liu et al., 2025). However, this two-stage approach has inherent limitations, as shown in Figures 17a to 17d. The fundamental problem is that clustering does not consider fitness during partitioning, which means that clusters may contain only weak individuals. When this occurs, even selecting the best model from such a cluster yields a poor representative, degrading ensemble performance. The Euclidean-based clustering illustrated in Figure 17b is further biased by uninformative diversity from poorly performing individuals located far from the main population, producing clusters dominated by low-quality models and reducing ensemble effectiveness. Using cosine distance as in Figure 17c alleviates this bias by focusing on angular relationships, yet the core limitation remains: regions containing only weak individuals still yield poor representatives. The key limitation is that clustering-based selection separates diversity modeling from quality assessment, preventing a proper balance between the two. In contrast, the DPP-based selection in Figure 17d jointly models both quality and diversity through a determinant-based kernel that couples fitness with semantic dissimilarity, enabling the construction of ensembles that are both diverse and high-performing.

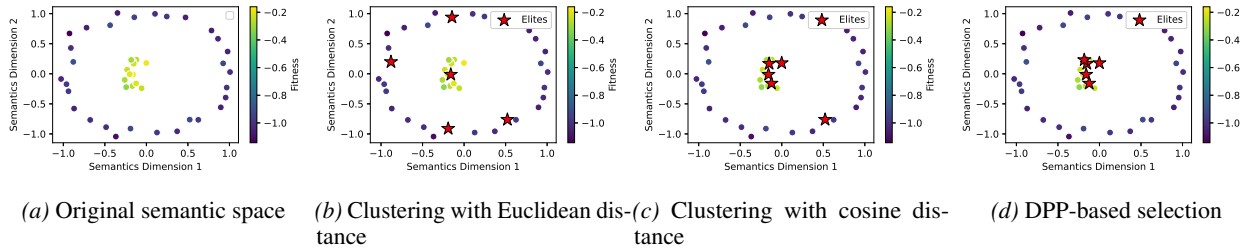

*(a)* Original semantic space  *(b)* Clustering with Euclidean dis-*(c)* Clustering with cosine dis-  *(d)* DPP-based selection
tance                          tance

*Figure 17.* Comparison of ensemble selection strategies on asymmetric semantic space. The clustering method used is K-Means.

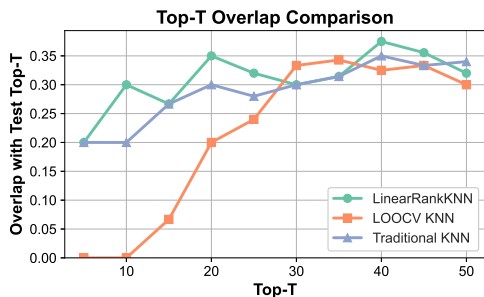

*Figure 18.* Comparison of linear-rank weighted KNN, LOOCV KNN, and traditional KNN fitness evaluation: overlap with true top-$T$ individuals ranked by test performance. Traditional KNN uses the same fixed $K$ as linear-rank weighted KNN but with uniform weighting.

## P. Advantages of Linear-Rank Weighted KNN for Feature Evaluation

The choice of evaluation metric during evolutionary feature construction critically influences which feature space is considered good. As discussed in Section 3.3.2, using adaptive LOOCV-based $K$ selection for fitness evaluation has limitations: different $K$ values for different individuals make the fitness score depend on both feature quality and $K$, rather than pure feature quality. To address this, we propose linear-rank weighted KNN regression for fitness evaluation, which uses a fixed $K$ and assigns weights $w_r = \frac{2(K-r+1)}{K(K+1)}$ to neighbors based on their rank $r$.

Figure 18 demonstrates the advantages of this approach, using individuals from the final population of Section 5.4. Linear-rank weighted KNN achieves higher overlap with true top-$T$ individuals ranked by test performance compared to both LOOCV KNN and traditional KNN, indicating more accurate identification of individuals that generalize well. The key difference lies in what each method optimizes. LOOCV KNN focuses primarily on prediction accuracy, whereas linear-rank weighted KNN evaluates representation quality by considering how smoothly the feature space transitions from similar to dissimilar samples. Traditional KNN, despite using the same fixed $k$ as linear-rank weighted KNN, performs worse because uniform weighting fails to capture the smoothness property of the feature space. This smoothness property is crucial for generalization, as it ensures that small changes in feature space correspond to gradual changes in target values. Linear-rank weighting naturally captures this smoothness by assigning weights proportional to neighbor rank, emphasizing closer neighbors while still considering the relative ordering of all neighbors. This provides a more faithful measure of representation quality, enabling more effective selection pressure during evolution and guiding the population toward feature constructions that yield better generalization.

## Q. Hyperparameter Optimization Details

All baseline methods are tuned on the training set only. For symbolic regression and classical machine learning baselines, we use the hyperparameter search spaces from SRBench (Cava et al., 2021). For the three modern deep tabular baselines, ModernNCA, MLP-PLR, and RealMLP, we use the search spaces in Table 7. We run heteroscedastic Bayesian optimization (HEBO) (Cowen-Rivers et al., 2022) for 32 evaluations per dataset. For tree-based methods, each candidate configuration is assessed using 5-fold cross-validation on the training set, and we select the configuration with the best average validation performance. For deep learning baselines, we further split the training set into 80% for training and 20% for validation during tuning to reduce computational cost. After hyperparameter selection, the final model is retrained on the full training set.

*Table 7.* Hyperparameter search spaces for deep learning baselines used in the optimization protocol in Appendix Q, adapted from (Ye et al., 2025). Each entry lists the variable type and its bounds. For hyperparameters prefixed with "?", the parameter is optional. We sample a boolean indicating whether it is enabled and, if enabled, sample its value within the given bounds.

| Algorithm | Hyperparameters |
|---|---|
| ModernNCA | Dropout: (uniform, 0.0, 0.5)
d_block: (int, 64, 1024)
n_blocks: (?int, 0, 0, 2)
dim: (int, 64, 1024)
num_embeddings: { n_frequencies: (int, 16, 96),
frequency_scale: (loguniform, 0.005, 10.0), d_embedding: (int, 16, 64) }
sample_rate: (uniform, 0.05, 0.6)
lr: (loguniform, 1e-05, 0.1)
weight_decay: (?loguniform, 0.0, 1e-06, 0.001) |
| MLP_PLR | d_layers: ($mlp\_d\_layers$, 1, 8, 64, 1024)
Dropout: (?uniform, 0.0, 0.0, 0.5)
num_embeddings: { n_frequencies: (int, 16, 96),
frequency_scale: (loguniform, 0.01, 100.0), d_embedding: (int, 16, 64) }
lr: (loguniform, 1e-05, 0.01)
weight_decay: (?loguniform, 0.0, 1e-06, 0.001) |
| RealMLP | num_emb_type: (categorical, [none, pbld, pl, plr])
add_front_scale: (categorical, [true, false])
lr: (loguniform, 0.02, 0.3)
p_drop: (categorical, [0.00, 0.15, 0.30])
act: (categorical, [selu, relu, mish])
hidden_sizes: (categorical, [[256, 256, 256], [64, 64, 64, 64, 64], [512]])
wd: (categorical, [0.0, 2e-2])
plr_sigma: (loguniform, 0.05, 0.5)
ls_eps: (categorical, [0.0, 0.1]) |

