# OpenReview forum: "Contrastive Symbolic Regression: Aligned Representations, Adaptive Prediction, and Diverse Ensembles"
_ICML.cc/2026/Conference — ICML 2026 regular_

### Official Review · Reviewer_YG9o · 2026-03-03

**Soundness:** 3
**Presentation:** 2
**Significance:** 4
**Originality:** 4
**Overall Recommendation:** 5
**Confidence:** 2

**Summary:**

This paper introduces a framework for contrastive symbolic regression. The main contribution of the work is adding a contrastive element to a symbolic regression workflow that leads to improved performance over other SOTA methods. The model transforms inputs into a feature space that organizes inputs such that inputs that produce similar outputs are grouped together. This allows for the use of K-NN regression on the constructed feature space which utilizes the organization of the space to its advantage to make predictions. The authors test their method on a variety of benchmark datasets against a variety of existing models and show that their method improves performance.

**Compliance With Llm Reviewing Policy:**

Affirmed.

**Final Justification:**

My final ruling for the paper is an accept. My initial review of the paper was strong, and the questions that I had for the authors were fully answered in the rebuttal period. As long as the figures that were mentioned in my initial review are indeed fixed for clarity, I think this paper will be a strong contribution to ICML.

**Key Questions For Authors:**

1. Could the authors quantify results in Figure 6? For example, calculate the variance of target values for each points K nearest neighbors and show that the CL objective lowers target variance of KNNs when compared to other formulations of the model.

**Limitations:**

yes

**Strengths And Weaknesses:**

### Strengths

1. The paper is very clear. The methods are very easy to follow and there is an extensive appendix for additional information. Additionally, the authors provide a very easy to read codebase to verify their results.
2. The authors propose a unique strategy for symbolic regression by employing contrastive learning in their model and doing so in an efficient way that does not necessitate gradient-based optimization.
3. The authors provide extensive benchmarking of their model against other SOTA methods and on a large collection of datasets.
4. The authors provide an extensive ablation study of their method, showing how different aspects of their design choices impact overall results.
5. The authors conduct statistical testing of their method compared to other methods.

### Weaknesses
1. Some of the figures in the paper are unclear. E.G. figure 8 and figure 4 report rank of methods but the captions say $R^2$, there are red lines in figure 7 heatmaps that are not explained in the caption, x-axis of figure 8 are miss-aligned, define what "model size" means in figure 5 middle, figure 2 is unclear.
2. It is not immediately clear from figure 6 that the contrastive learning method is grouping points by similarity of their output (see question 1).

---

> ### Author Rebuttal · Authors · 2026-03-29
>
> **W1 (Figure Clarity).** Thank you. We will revise the captions of Figures 4 and 8 to clarify that they report method rankings based on median test $R^2$. The red lines in Figure 7 highlight the region worth attention. We will also fix the misaligned x-axis in Figure 8. In Figure 5, "model size" refers to the number of nodes in the symbolic model, but in the final version, we will revise this definition to also include the weight matrix. With the weight matrix included, the overall model size will still be much lower than XGBoost. We will also revise Figure 2 to improve its clarity and provide more explanation in the figure.
>
> **Q1/W2 (Figure 6 Quantification).** Following your suggestion, we have quantified the effect of contrastive learning
> shown in Figure 6 by measuring the
> variance of target values within each sample's local neighborhood, defined by the nearest neighbors in feature
> space. The quantified results are Original = 0.097, FC = 0.109, and FC+CL = 0.054 with $K=5$. These results show
> that the contrastive objective reduces local target variance, making feature-space neighbors more
> target-consistent. We will include this statistic in the final revision.

---

> > ### Author Rebuttal · Reviewer_YG9o · 2026-04-01
> >
> > Thank you to the authors for responding to my questions. With the incorporated changes making the figures more detailed and adding statistics that show FC+CL is indeed grouping points by similarity, the authors have addressed all my concerns.

---

### Official Review · Reviewer_6a38 · 2026-03-11

**Soundness:** 3
**Presentation:** 3
**Significance:** 3
**Originality:** 3
**Overall Recommendation:** 4
**Confidence:** 2

**Summary:**

The paper introduces a symbolic regression method named Contrastive Symbolic Regression (CSR). CSR is an ensemble genetic programming method that leverages contrastive learning, DPP-based ensemble selection, and rank-weighted KNN. CSR is validated on a subset of the SRBench black-box dataset, where it outperforms all tested symbolic regression and machine learning methods. The paper conducts an extensive ablation study to validate the benefit of each component compared to other contemporary components.

**Compliance With Llm Reviewing Policy:**

Affirmed.

**Final Justification:**

The reviewer suggests accepting the paper. The rebuttal addresses my main concern by reclassifying the work as symbolic representation learning. The reclassification addresses my primary concern of interpretability, as symbolic regression has stricter interpretability requirements than symbolic representation learning. Additionally, the answer to Q1 significantly improved my perspective on the impact of CSR's methods. Overall, with the stipulations made during the rebuttal, the reviewer suggests accepting the paper.

**Key Questions For Authors:**

1. Can the authors provide the $R^2$ scores for the ablation study? While the ranking shows significance, the $R^2$ scores could be statistically similar. Additionally, the $R^2$ scores allow for comparison with Figure 5.
2. How does CSR perform on standard symbolic regression tasks? (Nguyen, Livermore, R, etc.)
3. Can CSR be considered a symbolic regression approach if it fails to "discover interpretable mathematical expressions"? The reviewer finds Appendix E to be insufficient to claim that CSR generates interpretable expressions.

The reviewer would be happy to significantly improve their score should the authors provide sufficient evidence that CSR can discover or recover **interpretable** mathematical expressions. The reviewer recognizes that question 2 is a significant request, but unless the authors can provide notable support for an exclusion of these experiments, the reviewer views it as a requirement for the soundness of the paper.

**Limitations:**

Yes.

**Strengths And Weaknesses:**

Strengths:
- Methodology: The method proposes a novel deterministic contrastive learning approach alongside two significant augmentations that enable strong predictive performance.
- Performance: CSR has strong performance on the evaluation dataset, achieving a higher $R^2$ score than ML-based and SR-based methods.

Weaknesses:
- Evaluation: The paper limits the evaluation of CSR to a subset of SRBench's black box problems. The evaluation highlights CSR's performance in this domain. Still, this test does not evaluate CSR's performance on standard symbolic regression problems, such as the Nguyen dataset or the symbolic recovery component of SRBench.
- Interpretability: CSR generates significantly complex expressions that result in the interpretability of the expression being lost. CSR generates expressions with complexity comparable to that of machine learning methods, thereby losing the interpretability expected in symbolic regression.

---

> ### Author Rebuttal · Authors · 2026-03-29
>
> **Q1 (Ablation $R^2$).** We agree that $R^2$ values are important. We provide the normalized median $R^2$ and
> bootstrapped 95% confidence
> intervals below. The median $R^2$ is normalized across algorithms on each dataset to account for different dataset
> difficulty levels and the resulting differences in $R^2$ ranges.
>
> These results show that the contrastive loss provides a clear benefit: the normalized median $R^2$ increases from 0.4613
> for SR$^\text{KNN}$ to 0.6064 for SR$^\text{CL+KNN}$.
>
> These ablation results are not directly comparable to those in Figure 5 because SRBench runs 10 runs, whereas the
> ablation study runs 30 runs with different random split settings.
>
> |                        | Normalized Median $R^2$ | 95% Confidence Interval |
> |:-----------------------|---------------------:|:------------------------|
> | SR$^\text{ALL}$        |               0.8884 | [0.7787, 0.9547]        |
> | SR$^\text{CL+LR+KNN}$  |                0.831 | [0.7528, 0.8579]        |
> | SR$^\text{PLS+LR+KNN}$ |               0.8244 | [0.6766, 0.8578]        |
> | SR$^\text{LR+KNN}$     |               0.8241 | [0.7120, 0.8967]        |
> | SR$^\text{LE+LR+KNN}$  |               0.7647 | [0.6756, 0.8392]        |
> | SR$^\text{LR+DT}$      |               0.7628 | [0.5499, 0.8615]        |
> | SR$^\text{LR}$         |               0.6602 | [0.2731, 0.7914]        |
> | SR$^\text{CL+LOO-KNN}$ |                0.641 | [0.5307, 0.7458]        |
> | SR$^\text{CL+KNN}$     |               0.6064 | [0.4541, 0.7437]        |
> | SR$^\text{LOO-KNN}$    |               0.5905 | [0.4353, 0.6900]        |
> | SR$^\text{DT}$         |               0.4861 | [0.3597, 0.7119]        |
> | SR$^\text{KNN}$        |               0.4613 | [0.3538, 0.6254]        |
>
> **Q2/W Evaluation (Standard SR Benchmarks).** We conducted experiments on the Nguyen, Livermore, and R benchmarks.
> Following the generation configuration in
> DSO [1], we generated the training and test sets and obtained the following results over 30 runs with different
> random seeds.
>
> |           | Median $R^2$ | 95% Confidence Interval |
> |:----------|----------:|:------------------------|
> | Livermore |    0.9999 | [0.9997, 1.0000]        |
> | Nguyen    |         1 | [0.9999, 1.0000]        |
> | R         |         1 | [1.0000, 1.0000]        |
>
> These results show that CSR performs consistently well on these benchmarks. In recent
> work, methods such as DySymNet [2] and MetaSymNet [3] also report very high $R^2$ values on Nguyen/Livermore/R.
>
> We mainly evaluate complex real-world regression tasks, where complex nonlinear relationships might not be well captured
> by a single parametric equation. In contrast, synthetic benchmarks such as Nguyen/Livermore/R are generated from
> predefined parametric equations and thus do not fully reflect the challenges targeted in this paper.
>
> **Q3/W Interpretability (Interpretability Claim).** CSR is closer to symbolic representation learning than to classical
> symbolic regression, and we will revise the paper
> to state this more explicitly. Modern symbolic regression has a broader scope. For example, SRBench, which is a
> comprehensive and standardized benchmarking framework for symbolic regression, also includes methods such as FEAT
> [4], MRGP [5], and ITEA [6], which construct symbolic features and then use a downstream linear predictor to make
> predictions.
> Regarding interpretability, we focus on feature-level explainability: the constructed features are symbolic
> expressions, and their influence can be analyzed through the learned coefficients and transformation weights. We will
> revise the paper to make the distinction between equation recovery and symbolic explainability clearer.
>
> [1]. "Deep symbolic regression: Recovering mathematical expressions from data via risk-seeking policy gradients." ICLR 2021.
> [2]. "A neural-guided dynamic symbolic network for exploring mathematical expressions from data." ICML 2024.
> [3]. "Metasymnet: A tree-like symbol network with adaptive architecture and activation functions." AAAI 2025.
> [4]. "Learning concise representations for regression by evolving networks of trees." ICLR 2019.
> [5]. "Multiple regression genetic programming." GECCO 2014.
> [6]. "Interaction-transformation evolutionary algorithm for symbolic regression." Evolutionary Computation 29.3, 2021:
> 367-390.

---

> > ### Author Rebuttal · Reviewer_6a38 · 2026-04-01
> >
> > The reviewer thanks the authors for their detailed response. The reviewer will increase their score significantly under the stipulations the reviewers provided.
> >
> > 1. The review agrees that there could be confusion about the comparison between these and Figure 5, but finds the results significantly more impactful for showcasing CSR's performance.
> > 2. The reviewer thanks the authors for running the additional experiments and recognizes that they are less crucial for symbolic representation learning than symbolic regression.
> > 3. The reviewer agrees with the key point that CSR is more akin to symbolic representation learning than to classical symbolic regression. The reviewer suggests expanding the related work section to capture the field of symbolic representation learning. The reviewer is more confident in the significance of this work with that classification and will update their score.

---

### Official Review · Reviewer_PYCu · 2026-03-12

**Soundness:** 3
**Presentation:** 3
**Significance:** 2
**Originality:** 2
**Overall Recommendation:** 4
**Confidence:** 4

**Summary:**

This paper introduces Contrastive Symbolic Regression (CSR), a new symbolic regression framework that combines genetic-programming-based feature construction with contrastive representation learning. Instead of only learning a direct input-to-target formula, CSR tries to build a symbolic feature space in which samples with similar target values are close to each other, and samples with very different targets are farther apart. In this way, the method turns symbolic regression partly into a representation learning problem, not just a formula fitting problem.

To make this practical within evolutionary search, the method derives a closed-form solution for a linear transformation of symbolic features, and then performs prediction with K-nearest-neighbor regression in the learned space. The framework further introduces efficient leave-one-out cross-validation for neighborhood-size selection, a linear-rank weighted KNN scheme for better representation assessment, and a DPP-based ensemble selection strategy to improve robustness by balancing model quality and diversity.

Empirically, the paper reports experiments on 58 real-world regression datasets and claims that CSR consistently outperforms both traditional symbolic regression baselines and several modern machine learning competitors. The overall message of the paper is that symbolic regression can be strengthened by learning geometry-aware symbolic representations, especially when prediction requires both interpretability and sensitivity to local neighborhood structure.

However, the method seems more likely to a representation method rather than a symbolic regression method. Lacks of related work discussion and discrimination make this issue further.

**Compliance With Llm Reviewing Policy:**

Affirmed.

**Final Justification:**

The additional clarifications are helpful and address several of my previous concerns. I will raise my evaluation accordingly.

**Key Questions For Authors:**

1. While the paper is framed as symbolic regression, the proposed method is more accurately viewed as a symbolic feature construction and representation learning framework combined with instance-based regression, since prediction is ultimately performed in a learned transformed feature space via KNN rather than through a direct symbolic expression. Could u justify the difference between this method to Feature Transformation [such as 1] and symbolic regression.

2. Will different distance function affect the model behavior?

3. The classification method and regression method should have different contrastive learning way. What is the difference between this two? Will discrete label make learning better or ?

4. You mainly optimize the  transformed feature distance and target distance matrix  rather than classic INFONCE. Would u further explain the core idea of this paper to contrastive learning?

5. How the interpretability works?

[1] FastFT: Accelerating Reinforced Feature Transformation via Advanced Exploration Strategies, ICDE 2025

**Limitations:**

Yes

**Strengths And Weaknesses:**

S1. The paper identifies a real limitation of prior symbolic regression methods
S2. Interesting hybridization of symbolic feature construction and representation learning.
S3. The experiments cover 58 real-world regression datasets, compare against both symbolic regression and non-symbolic tabular baselines, repeat runs over multiple seeds, and include significance testing. That gives the empirical section decent breadth

W1. The method is arguably closer to symbolic representation learning than to classical symbolic regression.
W2. Although the constructed features are symbolic, the final predictor consists of GP-generated features, a learned linear transform $W$, ridge boosting, KNN retrieval, and DPP ensembling. That is substantially less transparent than a direct symbolic expression, so the interpretability claim is only partial and mostly feature-level rather than predictor-level.
W3. Some methodological choices are not fully justified. Such as distance function.

---

> ### Author Rebuttal · Authors · 2026-03-29
>
> **Q1/W1 (SR vs Representation Learning).** We agree that CSR is better described as a symbolic feature construction and
> representation learning framework combined with instance-based regression.
> Compared with FastFT [1], which uses reinforcement learning to search for features, our approach uses evolutionary
> feature construction, but the key contributions include the contrastive objective, the KNN-based evaluation, and the
> ensemble aggregation strategy, all of which can also be combined with RL or other generators. Strictly speaking,
> symbolic regression is often associated with expression discovery. In practice, SRBench also includes methods that are
> closer to symbolic representation learning, such as FEAT [2], MRGP [3], and ITEA [4], since they still learn
> explainable symbolic transformations to predict a continuous target.
>
> **Q2/W3 (Distance Function).** Yes, different distance functions can affect model behavior in two ways: they change
> which samples are considered
> neighbors and how the transformation can be optimized. Since our predictor is KNN-based, altering the metric
> directly changes the neighborhood structure and can therefore change predictive behavior. We optimize Euclidean
> distance because it is the most commonly used KNN distance function and the resulting optimization has a closed-form
> solution. For alternative distances such as Chebyshev distance, the optimization no longer has a closed-form
> solution and typically requires iterative gradient-based optimization, which substantially increases computational
> cost.
>
> **Q3 (Regression vs Classification CL).** The contrastive learning objectives differ between regression and
> classification. For regression, we preserve
> finer-grained structure by matching pairwise target distances rather than grouping samples into "same" versus
> "different" classes. This is better aligned with continuous labels and more efficient, since Equation 7 gives
> a closed-form solution, which is important because symbolic feature construction must evaluate many candidate feature
> sets.
>
> **Q4 (Contrastive Objective vs InfoNCE).** The core idea of contrastive learning is to learn a representation whose
> pairwise geometry reflects target similarity. Instead of optimizing
> an InfoNCE objective built from binary positive/negative pairs, we directly match feature-space distances to
> continuous target distances. This is better aligned with regression, because nearby target values should map to
> nearby representations rather than being forced into a hard same/different grouping.
>
> **Q5/W2 (Explainability).** In CSR, explainability comes
> primarily from the symbolic feature layer. Each constructed feature is an explicit symbolic expression, so we can
> inspect the generated expressions. We then estimate feature importance by combining the ridge coefficients with the
> learned contrastive transformation matrix, which indicates which symbolic features most strongly influence the
> predictions. Therefore, CSR does not produce a single predictive equation, but it does provide feature-level
> explainability and traceability, as illustrated in Appendix E.
>
> [1] "FastFT: Accelerating Reinforced Feature Transformation via Advanced Exploration Strategies." ICDE 2025.
> [2]. "Learning concise representations for regression by evolving networks of trees." ICLR 2019.
> [3]. "Multiple regression genetic programming." GECCO 2014.
> [4]. "Interaction-transformation evolutionary algorithm for symbolic regression." Evolutionary Computation 29.3, 2021:
> 367-390.

---

> > ### Author Rebuttal · Reviewer_PYCu · 2026-04-04
> >
> > all my questions has been responded . But like the interpretability and methodology design still unclear. I think author needs to prepare a major revision at least

---

> > > ### Author Response · Authors · 2026-04-05
> > >
> > > Based on your suggestion, we have revised the paper accordingly. Specifically:
> > >
> > > * **Instance-level Interpretability.** We clarified at Line 46 of Section 1 that CSR can provide instance-level interpretability by retrieving similar historical instances in the learned symbolic feature space. This motivation was underemphasized previously. The revision is as follows:
> > >   *This alignment is particularly appealing in applications such as loan approval, where decisions should be justified by similar past instances. KNN can support a prediction by retrieving similar historical instances from the learned symbolic feature space, thereby providing instance-level interpretability.*
> > >
> > >   To illustrate this intuition, we ran SR$^\text{CL+LOO-KNN}$ on the U.S. crime-rate dataset. Here, `test` is the test instance and `nn1`-`nn5` are its five nearest neighbors in the learned symbolic feature space. For readability, the tables report original feature values.
> > >
> > >   |Role|$Y_{True}$|...|M|N|...|
> > >   |-|-|-|-|-|-|
> > >   |test|53.9||977|10||
> > >   |nn1|37.3||1045|6||
> > >   |nn2|66.4||986|22||
> > >   |nn3|54.2||968|4||
> > >   |nn4|74.2||984|34||
> > >   |nn5|65.3||948|113||
> > >
> > >   Using the original features gives the following neighbors.
> > >
> > >   |Role|$Y_{True}$|...|M|N|...|
> > >   |-|-|-|-|-|-|
> > >   |test|53.9||977|10||
> > >   |nn1|54.2||968|4||
> > >   |nn2|74.2||984|34||
> > >   |nn3|121.6||1018|10||
> > >   |nn4|127.2||964|9||
> > >   |nn5|34.2||965|6||
> > >
> > >   The learned symbolic space retrieves neighbors whose target values are closer to the test target than those retrieved from the original feature space. Because each data point corresponds to a U.S. state, the prediction can be explained to a decision maker through similar states with similar target values. This provides an instance-level interpretability advantage over classical parametric symbolic regression.
> > >
> > > * **Feature-level Interpretability.** We clarified at Line 818 in Appendix E that the proposed method provides feature-level interpretability via feature-importance analysis, consistent with FastFT [1]. The revision is as follows:
> > >   *This form of feature-level explainability is also used in feature transformation methods such as FastFT [1], which likewise analyze importance scores assigned to constructed features.*
> > >
> > > * For methodology design, we clarified three aspects: objective effectiveness, objective efficiency, and distance-function design.
> > >
> > >   **Objective Effectiveness.** We clarified at Line 72 of Section 1 why the proposed loss function is used instead of InfoNCE-style variants such as ordinal contrastive learning [2]. The revision is as follows:
> > >   *Ordinal contrastive formulations preserve ranked label relationships [2], but regression involves continuous supervision, and reducing targets to ordinal relations discards exact magnitude information. Therefore, it is desirable to learn a feature space whose geometry reflects continuous target proximity.*
> > >
> > >   **Objective Efficiency.** We clarified at Line 133 of Section 3.2.2 that the proposed objective yields a closed-form solution that is suitable for evolutionary feature construction. The revision is as follows:
> > >   *In contrast to ordinal contrastive losses that define ordered positive/negative relations and require gradient-based optimization [2], CSR preserves relationships based on continuous target proximity by directly matching pairwise target distances, which yields the closed-form solution in Equation (7).*
> > >
> > >   **Distance Function.** We clarified at Line 80 of Section 1 that a closed-form solution is important for efficiency in the evolutionary loop, which motivates the use of Euclidean distance because it admits a closed-form solution. The revision is as follows:
> > >   *Many contrastive objectives are optimized with iterative gradient-based methods; integrating such optimization within evolution is computationally expensive, motivating a closed-form solution.*
> > >
> > >   In addition, we would like to clarify that sensitivity to the distance function should be considered at two levels. At the lower level of KNN prediction, feature construction dynamically optimizes pairwise distances in the transformed feature space to improve prediction. At the higher level, changing the distance function in the contrastive objective may lead to different optimization results. Although KNN is well known to be sensitive to the distance function, the contrastive objective may be less sensitive. This is worth future investigation, especially because deriving closed-form solutions for other distance functions is non-trivial. To acknowledge this limitation, we added the following sentence after Line 439 in Section 6:
> > >   *Future work could also investigate alternative distance functions in CSR, although deriving closed-form solutions beyond Euclidean distance is non-trivial.*
> > >
> > > [1] "FastFT: Accelerating Reinforced Feature Transformation via Advanced Exploration Strategies." ICDE 2025.
> > > [2] "OCL: Ordinal Contrastive Learning for Imputating Features with Progressive Labels." MICCAI 2024.

---

### Official Review · Reviewer_VktY · 2026-03-15

**Soundness:** 3
**Presentation:** 2
**Significance:** 3
**Originality:** 3
**Overall Recommendation:** 4
**Confidence:** 3

**Summary:**

This paper proposes a contrastive symbolic regression framework. The main contributions of the paper include a closed-form solution for transforming GP-constructed features with contrastive learning, adaptive KNN, and determinantal point process-based ensemble selection. The authors evaluate the proposed approach on more than 50 regression tasks and show better accuracy with lower computational times compared to existing SR and MLP based approaches.

**Compliance With Llm Reviewing Policy:**

Affirmed.

**Key Questions For Authors:**

Questions:

1. Can you elaborate on the importance of relational structure importancnce in capturing local patterns and non-parametric variation (Line 091)
2. What is the y-axis in Fifure 4a top? The description says "test R^2", but the label says rank. Similar comment for Figure 8.
3. Figure 6 is a bit confusing. I understand that the PCA features for samples with similar target (closer in the colour scale) should be grouped together, that grouping improvement is not that clear from figure 6a to figure 6c.
4. .On line 051, it is mentioned that standard contrastive objective operates on binary similarity signal. However, there are several works on contrastive learning for ordinal regression and ordinal classification (Baek et al., Ordinal contrastive learning for imputating features with progressive labels; 	Saleem et al., Cloc: Contrastive learning for ordinal classification with multi-margin n-pair loss). I think it should be mentioned in the related work.
5. Line 182 says " This LOOCV-based selection, however, has limitations when evaluating feature space quality.", can you provide more explanation for this?
6. In the evaluation section, I could not find any ablation study for the impact of DPP. How much difference does it make?



Minor:

1. The x-axis labels in Figure 4 are a bit hard to read.
2. Figure 7 trend is also a bit difficult to understand. It would be better to understand if the samples are sorted by target value. We would expect the values further from the diagonal to be darker if the feature space preserves the distance of the target space.
3. Linear regression boosting is introduced directly in Section 3. Similarly noveltiesfor KNN-based evaluation are directly mentioned in the contributions at the end of the introduction. The writing and introduction of the ideas can be improved.
The structure of Section 3 is a bit confusing. Section 3.2 is named "Contrastive Symbolic Regression", but then other components, like the transformation matrix, are in a different subsection, 3.3.

**Limitations:**

The evaluation is limited to single-modality small-scale regression tasks (as noted by the authors as well). It would be good to have a bit more discussion on that- challenges associated with larger tasks, computational time, and accuracy trade-offs compared to other approaches.

**Strengths And Weaknesses:**

Strengths:

1. The paper tackles an important problem of improving accuracy of symbolic regression while keeping computational cost low and provides a novel solution of using closed form contrasitve learning solution.
2. Ideas are well presented and organized, making it easy to follow the paper. I think most components introduced in the paper are well explained and motivated.
3. Evaluation is thorough. The evaluation includes ablation studies to highlight the importance of each component.


Weakness:

1. Some of the figures are a bit difficult to understand.
2. Evaluation limited to smaller regression tasks
3. Paper writing and organization can be improved. Some of the contributions/ideas are directly introduced in Section 3. Section 3's subsection structure is a bit confusing.  See comments/questions below.

---

> ### Author Rebuttal · Authors · 2026-03-29
>
> **Major**
>
> **Q1 (Relational Structure).** Relational structure encodes how instances behave locally, allowing models to capture
> smooth, region-specific
> patterns even when global relationships are complex. It lays the foundation for capturing non-parametric variation by
> avoiding fixed
> functional assumptions and enabling flexible adaptation across different regions of the input space.
>
> **Q2/W1 (Figure Rank Axis).** The y-axis values in Figure 4 (top) and Figure 8 (left) report the average rank of methods
> based on test $R^2$
> across datasets.
> We will revise the captions to make this clearer.
>
> **Q3/W1 (Figure 6 Clarity).** To show the grouping improvement more clearly, we quantify it by computing the variance of
> target values among each
> point's 5 nearest neighbors according to Reviewer YG9o's suggestion. The variances are Original = 0.097, FC = 0.109, and
> FC+CL = 0.054. These results
> indicate that contrastive learning makes feature-space neighbors more target-consistent.
>
> **Q4 (Ordinal CL Related Work).** Thank you. We will add references on ordinal contrastive learning. The main difference
> is that our method preserves
> relationships based on continuous target proximity rather than ordinal positive/negative pairs. This formulation has
> a closed-form solution and is therefore more efficient than Ordinal Contrastive Loss [1], which relies on
> gradient-based optimization.
>
> **Q5 (LOOCV Limitation).** The limitation of LOOCV-based $K$ selection is that it evaluates a predictor and a
> hyperparameter jointly, so the
> LOOCV score mixes representation quality with the choice of $K$. Two similar feature spaces may therefore receive
> different scores mainly because they select different
> neighborhood sizes. For example, one feature space may select $K=2$, while another selects $K=10$ based on LOOCV. These
> LOOCV scores are not directly comparable because
> the choice of $K$ heavily influences prediction quality. This is why we use fixed-$K$ linear-rank weighted KNN during
> evolution and reserve LOOCV for final
> model selection. We have provided an example in Appendix P showing that LOOCV may fail to assign correct fitness to
> feature
> spaces that generalize well on the test set.
>
> **Q6 (DPP Ablation).** Figure 8 presents the ablation study results, which show that the DPP-selected ensemble
> outperforms top-model
> selection. DPP mainly improves robustness by balancing accuracy and diversity in the selected ensemble. If we only
> keep the top models, they tend to be similar and therefore make correlated errors, thus limiting the benefit of
> averaging. DPP instead favors accurate but non-redundant models, which improves ensemble performance.
>
> **Minor**
>
> **Q1/W1 (Figure 4 Labels).** We will enlarge the x-axis labels in Figure 4.
>
> **Q2/W1 (Figure 7 Clarity).** We will revise Figure 7 accordingly to improve its clarity.
>
> **Q3/W3 (Paper Organization).** We will introduce linear regression boosting before Section 3 to clarify that linear
> regression is used to capture
> the global trend. The motivation for efficient LOOCV KNN and rank-weighted KNN evaluation is introduced
> around line 86. We will also present the contrastive objective and the
> transformation matrix more coherently by merging related paragraphs.
>
> **Weakness**
>
> **W2 (Evaluation Scope).** We agree that the current evaluation is centered on small-to-medium tabular regression tasks.
> This focus is intentional because CSR is designed for data-scarce tabular problems, where feature construction is
> especially useful [2][3], and recent strong baselines such as RealMLP [4] and ModernNCA [5] also emphasize improving
> performance on small tabular datasets. A broader large-scale evaluation is left to future work.
>
> [1]. "OCL: Ordinal Contrastive Learning for Imputating Features with Progressive Labels." MICCAI 2024.
> [2]. "An extreme learning machine based virtual sample generation method with feature engineering for credit risk
> assessment with data scarcity." Expert Systems with Applications 202 (2022): 117363.
> [3]. "Expanded feature space-based gradient boosting ensemble learning for risk prediction of type 2 diabetes
> complications." Applied Soft Computing 144 (2023): 110451.
> [4]. "Better by default: Strong pre-tuned mlps and boosted trees on tabular data." NeurIPS 2024.
> [5]. "Revisiting Nearest Neighbor for Tabular Data: A Deep Tabular Baseline Two Decades Later." ICLR 2025.

---

> > ### Author Rebuttal · Reviewer_VktY · 2026-04-03
> >
> > I would like to thank the authors for the rebuttal. The authors' response addresses all my concerns.

---

### Decision · Program_Chairs · 2026-04-30

**Decision:**

Accept (regular)

**Comment:**

All four reviewers acknowledged the author rebuttals and shared positive ratings (three WAs and one A).
This work proposes an interpretable regression mode based on GP-constructed feature, named CSR (Contrastive Symbolic  Regression), and a rank-weighted kNN-based evaluation for a representation assessment. Reviewers indicated that this work identifies a real limitation of existing SR methods and valued their novel contrastive learning approach.

Reviewers initially shared common concerns about its presentation and evaluation.
The presentation concern seems resolved through the rebuttal process, according to the reviewers. I expect the authors to make the corresponding changes in the next revision.

Reviewers VktY and 6a38 pointed out limited evaluations in this study. The initial experiment considered black-box problems only. The authors provided additional experiments using symbolic regression datasets (Livermore, Nguyen, and R). Yet, those are relatively simple datasets and I do not see the addition as a significant contribution, but rather find the concern sufficiently addressed yet.

Overall, I think this work is worth discussing at ICML'26 if its space allows.